# COPULA CONFORMAL PREDICTION FOR MULTI-STEP TIME SERIES FORECASTING

**Sophia Sun**
University of California, San Diego
shs066@ucsd.edu

**Rose Yu**
University of California, San Diego
roseyu@ucsd.edu

## ABSTRACT

Accurate uncertainty measurement is a key step in building robust and reliable machine learning systems. Conformal prediction is a distribution-free uncertainty quantification framework popular for its ease of implementation, finite-sample coverage guarantees, and generality for underlying prediction algorithms. However, existing conformal prediction approaches for time series are limited to single-step prediction without considering the temporal dependency. In this paper, we propose the **Copula C**onformal **P**rediction algorithm for multivariate, multi-step **T**ime **S**eries forecasting, **CopulaCPTS**. We prove that CopulaCPTS has finite-sample validity guarantee. On four synthetic and real-world multivariate time series datasets, we show that CopulaCPTS produces more calibrated and efficient confidence intervals for multi-step prediction tasks than existing techniques. Our code is open-sourced at https://github.com/Rose-STL-Lab/CopulaCPTS.

## 1 INTRODUCTION

Deep learning models are becoming widely used in high-risk settings such as healthcare and transportation. In these settings, it is important that a model produces calibrated uncertainty to reflect its own confidence. Confidence regions are a common approach to quantify prediction uncertainty (Khosravi et al., 2011). A $(1 - \alpha)$-confidence region $\Gamma^{1-\alpha}$ for a random variable $y$ is *valid* if it contains $y$'s true value with high probability: $\mathbb{P}[y \in \Gamma^{1-\alpha}] \geq 1 - \alpha$. Note that one can make the confidence region infinitely large to satisfy validity. But for the confidence region to be useful, we also want to minimize its area while remaining valid; this is known as the *efficiency* of the region.

Conformal prediction (CP) is a powerful framework to produce confidence regions with finite-sample guarantees of validity (Vovk et al., 2005; Lei et al., 2018). Furthermore, it makes no assumptions about the underlying prediction model or the data distribution. CP's generality, simplicity, and statistical guarantees have made it popular for many real-world applications including time series prediction (Xu & Xie, 2021), drug discovery (Eklund et al., 2015) and safe robotics (Luo et al., 2021).

This paper considers the setting of multi-step time series forecasting from a set of independent sequences. Consider the problem of vehicle trajectory prediction, illustrated in Figure 1. Given a

Time steps: temporal dependence (non-i.i.d.)      Dataset $\mathscr{D}$: independent observations (i.i.d.)

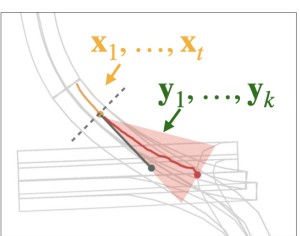
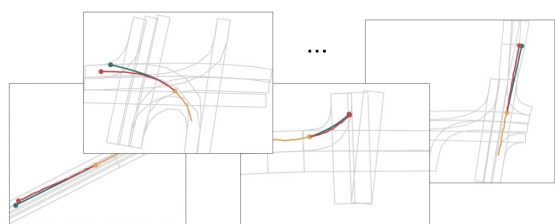

Figure 1: Illustration of the multi-step time series forecasting setting. (Left) The timesteps within a time series are temporally dependent, and (Right) the observations in the dataset are independent.

dataset of trajectories, the task is to predict a future trajectory for $k$ steps given its past trajectory of $t$ time steps. We assume that the trajectories are independent from each other. For each trajectory, these time steps are temporally dependent.

There are many real-world tasks that present the same challenges as the example above, such as EEG forecasting (each patient is independent), short-term weather forecasting (local meteorology history is independent), etc. They require predicting multiple time steps into the future, so it is desired to have a "cone of uncertainty" that covers the entire course of the forecasts. Existing CP methods for time series data either only provide coverage guarantee for individual time steps (Gibbs & Candes, 2021; Xu & Xie, 2021) or produce confidence regions that are often too inefficient to be useful, especially in long horizons or multivariate settings (Stankevičiūtė et al., 2021).

In this paper, we present a practical and effective conformal prediction algorithm for multi-step time series forecasting. We introduce **CopulaCPTS**, a **Copula**-based **C**onformal **P**rediction algorithm for multi-step **T**ime **S**eries forecasting. A copula is a multivariate cumulative distribution function that models the dependence between multiple random variables. By using copulas to model the uncertainty jointly over future time steps, we can shrink the confidence regions significantly while maintaining validity. Copulas have been used for conformal prediction (Messoudi et al., 2021), but they focus on multiple target prediction in non-temporal settings and did not provide a validity proof.

In summary, our contributions are:

- We introduce **CopulaCPTS**, a general uncertainty quantification algorithm that can be applied to *any* multivariate multi-step forecaster.

- We prove that **CopulaCPTS** produces valid confidence regions for the full forecast horizon.

- **CopulaCPTS** produces significantly sharper and more calibrated uncertainty estimates than state-of-the-art baselines on two synthetic and two real-world benchmark datasets.

- We extend **CopulaCPTS** to obtain valid confidence intervals for time series forecasts of varying lengths.

## 2 RELATED WORK

**Deep Uncertainty Quantification for Time-Series Forecasting.** The two major paradigms of Uncertainty Quantification (UQ) methods for deep neural networks are Bayesian and Frequentist. Bayesian approaches estimate a distribution over the model parameters given data, and then marginalize these parameters to form output distributions via Markov Chain Monte Carlo (MCMC) sampling (Welling & Teh, 2011; Neal, 2012; Chen et al., 2014) or variational inference (VI) (Graves, 2011; Kingma et al., 2015; Blundell et al., 2015; Louizos & Welling, 2017). Wang et al. (2019); Wu et al. (2021) propose Bayesian Neural Networks (BNN) for UQ of spatiotemporal forecasts. In practice, Bayesian UQ can be computationally expensive and difficult to optimize, especially for larger networks (Lakshminarayanan et al., 2017; Zadrozny & Elkan, 2001). Furthermore, Bayesian methods do not provide any finite sample coverage guarantees. Therefore, UQ for deep neural network time series forecasts often adopts approximate Bayesian inference such as MC-dropout (Gal & Ghahramani, 2016b; Gal et al., 2017).

Frequentist UQ methods emphasize robustness against variations in the data. These approaches either rely on resampling the data or learning an interval bound to encompass the dataset. For time series forecasting UQ, approaches include ensemble methods such as bootstrap (Efron & Hastie, 2016; Alaa & Van Der Schaar, 2020) and jackknife methods (Kim et al., 2020; Alaa & Van Der Schaar, 2020); interval prediction methods include interval regression through proper scoring rules (Kivaranovic et al., 2020; Wu et al., 2021), and quantile regression (Takeuchi et al., 2006), with many recent advances for time series UQ (Tagasovska & Lopez-Paz, 2019; Gasthaus et al., 2019; Park et al., 2022; Kan et al., 2022). Many of the frequentist methods produce asymptotically valid confidence regions and can be categorized as distribution-free UQ techniques as they are (1) agnostic to the underlying model and (2) agnostic to data distribution.

**Conformal Prediction.** Conformal prediction (CP) is an important member of distribution-free UQ methods; we refer readers to Angelopoulos & Bates (2021) for a comprehensive introduction and survey of CP. CP has become popular because of its simplicity, generality, theoretical soundness, and

low computational cost. A key feature of CP is that under the exchangeability assumption, conformal methods guarantee validity in finite samples (Vovk et al., 2005).

Most relevant to our work is the recent endeavor in generalizing CP to time-series forecasting. According to Stankevičiūtė et al. (2021) there are two settings: data generated from (1) one single time series or (2) multiple independent time series. For the first setting, ACI (Gibbs & Candes, 2021) and EnbPI (Xu & Xie, 2021) developed CP algorithms that relax the exchangeability assumption while maintaining asymptotic validity via online learning (former) and ensembling (later); Zaffran et al. (2022) further improves online adaptation. Sousa et al. (2022) combines EnbPI with conformal quantile regression (Romano et al., 2019) to model heteroscedastic time series. However, because these algorithms operate on one single time series, the validity guarantees do not cover the full horizon, posing issues for application in high-risk settings.

We focus on the setting where data consists of many independent time series. Stankevičiūtė et al. (2021) shares the same setting as ours but provides only a univariate time series solution. We show that their method of applying Bonferroni correction produces inefficient confidence regions, especially for multidimensional data or long prediction horizons. Messoudi et al. (2021) uses a copula function for multi-target CP for non-temporal data, creating box-like regions to account for the correlations between the labels. However, they do not provide theoretical proof and empirical results have shown that are often invalid. This paper builds upon these works and presents a novel two-step algorithm with guaranteed multivariate multi-step coverage and efficient confidence regions.

## 3 BACKGROUND

### 3.1 INDUCTIVE CONFORMAL PREDICTION (ICP)

Let $\mathcal{D} = \{z^i = (x^i, y^i)\}_{i=1}^n$ be a dataset with input $x^i \in \mathcal{X}$ and output $y^i \in \mathcal{Y}$ such that each data point $z^i \in \mathcal{Z} := \mathcal{X} \times \mathcal{Y}$ is drawn i.i.d. from an unknown distribution $\mathcal{Z}$.

We will briefly present the algorithm and theoretical results for conformal prediction, and refer readers to Angelopoulos & Bates (2021) for a thorough introduction. The goal of conformal prediction is to produce a *valid* confidence region (Def. 3.1) for any underlying prediction model.

**Definition 3.1** (Validity). Given a new data point $(x, y)$ and a desired confidence $1 - \alpha \in (0, 1)$, the confidence region $\Gamma^{1-\alpha}(x)$ is a subset of $\mathcal{Y}$ containing probable outputs $\tilde{y} \in \mathcal{Y}$ given $x$. The region $\Gamma^{1-\alpha}$ is valid if

$$\mathbb{P}[y \in \Gamma^{1-\alpha}(x)] \geq 1 - \alpha \tag{1}$$

Conformal prediction splits the dataset into a proper training set $\mathcal{D}_{train}$ and a calibration set $\mathcal{D}_{cal}$. A prediction model $\hat{f} : \mathcal{X} \to \mathcal{Y}$ is trained on $\mathcal{D}_{train}$. We use a *nonconformity score* $A : \mathcal{Z}^{|\mathcal{D}_{train}|} \times \mathcal{Z} \to \mathbb{R}$ to quantify how well a data sample from calibration *conforms* to the training dataset. Typically, we choose a metric of disagreement between the prediction and the true label as the non-conformity score, such as the Euclidean distance:

$$A(\mathcal{D}_{train}, (x, y)) \stackrel{\text{e.g.}}{=} d(y, \hat{f}(x)) \stackrel{\text{e.g.}}{=} \|y - \hat{f}(x)\|_2 \tag{2}$$

For conciseness, we write $A(\mathcal{D}_{train}, (x^i, y^i))$ as $A(z^i)$ in rest of the paper.

Let $\mathcal{S} = \{A(z^i)\}_{z^i \in \mathcal{D}_{cal}}$ denote the set of nonconformity scores of all samples in the calibration set $\mathcal{D}_{cal}$. We can define a quantile function for the nonconformity scores $\mathcal{S}$ as:

$$Q(1 - \alpha, \mathcal{S}) := \inf\{s^* : (\frac{1}{|\mathcal{S}|} \sum_{s^i \in \mathcal{S}} \mathbb{1}_{s^i \leq s^*}) \geq 1 - \alpha\}. \tag{3}$$

Conformal prediction is guaranteed to produce valid confidence regions (Vovk et al., 2005), under the exchangeablility assumption defined as follows,

**Definition 3.2** (Exchangeability). In a dataset $\{z^i\}_{i=1}^n$ of size $n$, any of its $n!$ permutations are equally probable.

The procedure introduced above is known as *inductive* conformal prediction, as it splits the dataset into training and calibration sets to reduce computation load (Vovk et al., 2005; Lei & Wasserman, 2012). Our method is based on inductive CP, but can also be easily adapted for other CP variants.

## 3.2 COPULA AND ITS PROPERTIES

Copula is a concept from statistics that describes the dependency structure in a multivariate distribution. It has also been used in generative models for multivariate time series (Salinas et al., 2019; Drouin et al., 2022). We can use copulas to capture the joint distribution for multiple future time steps. We briefly introduce its notations and concepts.

**Definition 3.3** (Copula). Given a random vector $(X_1, \cdots X_k)$, define the marginal cumulative density function (CDF) for each variable $X_h, h \in \{1, \ldots, k\}$ as

$$F_h(x) = \mathbb{P}[X_h \leq x]$$

The copula of $(X_1, \cdots X_k)$ is the joint CDF of $(F_1(X_1), \cdots, F_k(X_k))$, written as

$$C(u_1, \cdots, u_k) = \mathbb{P}\left[F_1(X_1) \leq u_1, \cdots, F_t(X_k) \leq u_k\right]$$

In other words, the copula function captures the dependency structure between the variable $X$s; we can view an $k$ dimensional copula $C : [0, 1]^k \to [0, 1]$ as a CDF with uniform marginals, as illustrated in Figure 2. A fundamental result in the theory of copula is Sklar's theorem.

**Theorem 3.4** (Sklar's theorem). *Given a joint CDF as $F(X_1, \cdots, X_k)$ and the marginals $F_1(x), \ldots, F_k(x)$, there exists a copula such that*

$$F(x_1, \cdots, x_k) = C(F_1(x_1), \cdots, F_k(x_k))$$

*for all $x_j \in (-\infty, \infty), j \in \{1, \ldots, k\}$.*

Sklar's theorem states that for all multivariate distribution functions, there exists a copula function such that the distribution can be expressed using the copula and multiple univariate marginal distributions. When all the $X_k$s are independent, the copula function is known as the product copula: $C(u_1, \cdots, u_k) = \Pi_{i=1}^k u_i$.

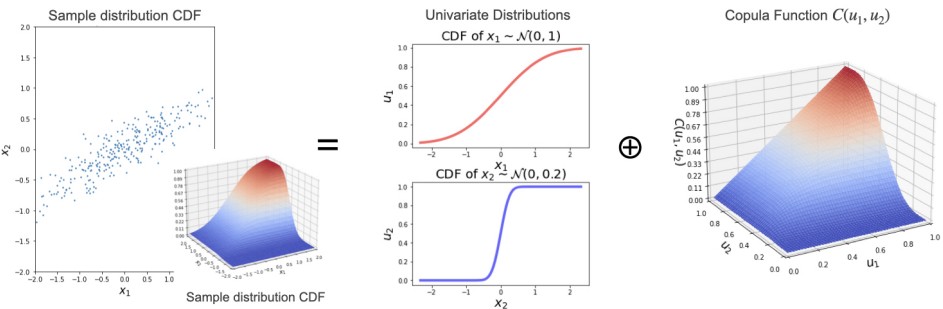

Figure 2: An example copula, where we express a multivariate Gaussian with correlation $\rho = 0.8$ with two univariate distributions and a copula function $C(u_1, u_2)$.

## 4 COPULA CONFORMAL PREDICTION FOR TIME SERIES (COPULACPTS)

UQ methods are evaluated on two properties: *validity* and *efficiency*. A model is *valid* when the predicted confidence is greater than or equal to the probability of events falling into the predicted range (Definition 3.1). The term *calibration* describes the case of equality in the validity condition. *Efficiency*, on the other hand, refers to the size of the confidence region. In practice, we want the measure of the confidence region (e.g. its area or length) to be as small as possible, given that the validity condition holds. CopulaCPTS improves the efficiency of confidence regions by modeling the dependency of the time steps using a copula function.

Denote the time series dataset of size $n$ as $\mathcal{D} = \{z^i = (x_{1:t}^i, y_{1:k}^i)\}_{i=1}^n$, where $x_{1:t} \in \mathbb{R}^{t \times d}$ is $t$ input steps, and $y_{1:k} \in \mathbb{R}^{k \times d}$ is $k$ prediction steps, both with dimension $d$ at each step. Each data point $z^i$ is sampled i.i.d. from an unknown distribution $\mathcal{Z}$. In the multi-step forecasting setting, given a confidence level $1 - \alpha$, the algorithm produces $k$ confidence regions for a test input $x_{1:t}^{n+1}$, denoted as $[\Gamma_1^{1-\alpha}, \ldots, \Gamma_k^{1-\alpha}]$. We say the confidence regions are *valid* if all time steps in the forecast are covered:

$$\mathbb{P}[\forall j \in \{1, \ldots, k\}, \ y_j \in \Gamma_j^{1-\alpha}] \geq 1 - \alpha. \tag{4}$$

In the following sections, we introduce CopulaCPTS, a conformal prediction algorithm that is both valid and efficient for multivariate multi-step time series forecasts.

## 4.1 Algorithm Details

The key insight of our algorithm is that we can model the joint probability of uncertainty for multiple predicted time steps with a copula, hence better capturing the confidence regions. We divide the calibration set $\mathcal{D}_{cal}$ into two subsets: $\mathcal{D}_{cal-1}$, which we use to estimate a Cumulative Distribution Function (CDF) on the nonconformity score of each time step, and $\mathcal{D}_{cal-2}$, to calibrate the copula.

The two calibration sets allow us to prove validity for both components of our algorithm. At the cost of using a subset of the data to calibrate a copula, our approach produces provably more efficient confidence regions compared to worst-case corrections such as union bounding in Stankevičiūtė et al. (2021) which is a lower bound for copulas (Appendix B.1), and more valid regions than Messoudi et al. (2021) (table 1).

**Nonconformity of Multivariate Forecasts.** If the time series is multivariate, we have each target time step $y_j \in \mathbb{R}^d$. Given $z = (x, y) \sim \mathcal{Z}$, let the nonconformity score be the L-2 distance $s_j^i = A(z^i)_j \overset{\text{e.g.}}{=} \|y_j^i - \hat{f}(x^i)_j\|$ for each timestep $j = 1, \ldots, k$, where $\hat{f}(x)$ is a forecasting model trained on $\mathcal{D}_{train}$. The confidence region $\Gamma^{1-\alpha}(x)$ therefore is a $d$-dimensional ball. We chose this metric for simplicity, but one can choose other metrics such as Mahalanobis (Johnstone & Cox, 2021) or L-1 (Messoudi et al., 2021) distance based on domain needs, and our algorithm will remain valid.

For brevity, we will use $\mathcal{S}1 = \{s^i\}_{z^i \in \mathcal{D}_{cal-1}}$ to denote the set of nonconformity scores of data in $\mathcal{D}_{cal-1}$ and $\mathcal{S}2 = \{s^i\}_{z^i \in \mathcal{D}_{cal-2}}$ the set of nonconformity scores of data in $\mathcal{D}_{cal-2}$. Subscript $j$ will be used to index the set of specific time steps of the scores: $\mathcal{S}1_j = \{s_j^i\}_{z^i \in \mathcal{D}_{cal-1}}$, $\mathcal{S}2_j = \{s_j^i\}_{z^i \in \mathcal{D}_{cal-2}}$.

**Calibrating CDF on $\mathcal{D}_{cal-1}$.** We use $\mathcal{D}_{cal-1}$ to build conformal predictive distributions for each time step's anomaly scores, which provides desirable validity properties (Vovk et al., 2017). The conformal cumulative distribution function is constructed as follows. [1]

$$\hat{F}_j(s_j) := \frac{1}{|\mathcal{S}1_j| + 1}\Big(\tau + \sum_{s^i \in \mathcal{S}1_j} \mathbb{1}_{s_j^i < s_j}\Big), \text{ where } \tau \sim (0, 1), \text{ for } j \in \{1, \ldots, k\} \tag{5}$$

**Copula Calibration on $\mathcal{D}_{cal-2}$.** Next, for every data point in $\mathcal{D}_{cal-2}$, we evaluate the cumulative probability of its anomaly scores with the estimated conformal predictive distributions:

$$\mathcal{U} = \{\mathbf{u}^i\}_{i \in \mathcal{D}_{cal-2}}, \quad \mathbf{u}^i = (u_1^i, \ldots, u_k^i) = \big(\hat{F}_1(s_1^i), \ldots, \hat{F}_k(s_k^i)\big) \tag{6}$$

We adopt the empirical copula (Ruschendorf, 1976) for modeling and proof in this work. The empirical copula is a non-parametric method of estimating marginals directly from observation, and hence does not introduce any bias. For the joint distribution of $k$ time steps, we construct $C_{\text{empirical}} : [0, 1]^k \to [0, 1]$ as Eqn 7.

$$C_{\text{empirical}}(\mathbf{u}) = \frac{1}{|\mathcal{D}_{cal-2}| + 1} \sum_{i \in \mathcal{D}_{cal-2} \cup \{\boldsymbol{\infty}\}} \prod_{j=1}^{k} \mathbb{1}_{\mathbf{u}_j^i < \mathbf{u}_j} \tag{7}$$

Here boldface $\boldsymbol{\infty}$ is a k-dimensional vector with each $\boldsymbol{\infty}_j = \infty$ for $j = 1, \ldots, k$.

To fulfill the full-horizon validity condition of Eqn 4, we only need to find appropriate $\mathbf{u}^*$ such that $C_{\text{empirical}}(\mathbf{u}^*) \geq 1 - \alpha$.

$$\underset{\mathbf{u}^*}{\arg\min} \sum_{j=1}^{k} \mathbf{u}_j^* \quad \text{s.t. } C_{\text{empirical}}(\mathbf{u}^*) \geq 1 - \alpha \tag{8}$$

---

[1] Because of the random component, equation 5 is a "thick" CDF of thickness $\frac{1}{|\mathcal{S}1|+1}$, which becomes inconsequential when the calibration set is large. See Vovk et al. (2017) for theoretical justifications. In implementation, we treat $\tau = 1$ for simplicity and better coverage.

Note that the $\mathbf{u}^*$ is not and does not have to be unique; any solution that satisfies the constraint in Eq. 8 will guarantee multi-step validity (Appendix A). The minimization helps with efficiency, i.e. the sharpness of the confidence regions. We use a gradient descent algorithm for the optimization in implementation (see Appendix B.2 for details, and Appendix C.5 for a study of its effectiveness). Lastly, We obtain $(s_1^*, \ldots, s_k^*)$ by $\hat{F}_j^{-1}(\mathbf{u}_j^*)$ and construct the confidence region for each time step $j \in \{1, \ldots, k\}$ as the set of all $y_j \in \mathbb{R}^d$ such that the nonconformity score is less than $s_j^*$. Algorithm 1 summarizes the CoupulaCPTS procedure.

The full proof of CopulaCPTS's validity (theorem 4.1) can be found in Appendix A. Intuitively, CopulaCPTS performs conformal prediction twice: first calibrating the nonconformity scores of each time step with $\mathcal{D}_{cal-1}$, and then calibrating the copula with $\mathcal{D}_{cal-2}$.

**Theorem 4.1** (Validity of CopulaCPTS). *CopulaCPTS (algorithm 1) produces valid confidence regions for the entire forecast horizon. i.e.*

$$\mathbb{P}[\forall j \in \{1, \ldots, k\}, y_j \in \Gamma_j^{1-\alpha}] \geq 1 - \alpha.$$

---

**Algorithm 1:** Copula Conformal Time Series Prediction (**CopulaCPTS**)

**Input:** Dataset $\mathcal{D}$, test inputs $\mathcal{D}_{test}$, target significant level $1 - \alpha$.
**Output:** $\Gamma_1^{1-\alpha}, \ldots, \Gamma_k^{1-\alpha}$ for each test input.

1 ———————————————————
2 // Training
3 Randomly split dataset $\mathcal{D}$ into $\mathcal{D}_{train}$ and $\mathcal{D}_{cal}$.
4 Train $k$-step forecasting model $\hat{f}$ on training set $\mathcal{D}_{train}$.
5 // Calibration
6 Randomly split $\mathcal{D}_{cal}$ into $\mathcal{D}_{cal-1}$ and $\mathcal{D}_{cal-2}$.
7 **for** $(x_{1:t}^i, y_{1:k}^i) \in \mathcal{D}_{cal-1} \cup \mathcal{D}_{cal-2}$ **do**
8 $\quad \hat{y}_{1:k}^i \leftarrow \hat{f}(x_{1:t}^i)$
9 $\quad s_j^i \leftarrow \|y_j^i - \hat{y}_j^i\|$ **for** $j = 1, \ldots, k$
10 **end for**
11 $\hat{F}_1, \ldots, \hat{F}_k \leftarrow$ Eq. (5) on $\mathcal{D}_{cal-1}$
12 $C_{\text{empirical}}(\cdot) \leftarrow$ Eq. (7) on $\mathcal{D}_{cal-2}$
13 $\mathbf{u}^* \leftarrow$ Eq. (8)
14 $s_j^* = \hat{F}_j^{-1}(\mathbf{u}_j^*)$ **for** $j = 1, \ldots, k$
15 // Prediction
16 **for** $x_{1:t}^i \in \mathcal{D}_{test}$ **do**
17 $\quad \hat{y}_{1:k}^i \leftarrow \hat{f}(x_{1:t}^i)$
18 $\quad \Gamma_j^{1-\alpha} \leftarrow \{y : \|y - \hat{y}_h^i\| < s_j^*\}$ **for** $j = 1, \ldots, k$
19 $\quad$ **yield** $\Gamma_1^{1-\alpha}, \ldots, \Gamma_k^{1-\alpha}$
20 **end for**

---

## 5 EXPERIMENTS

In this section, we show that CopulaCPTS produces more calibrated and efficient confidence regions compared to existing methods on two synthetic datasets and two real-world datasets. We demonstrate that CopulaCPTS's advantage is more evident over longer prediction horizons in Section 5.2. We also show its effectiveness in the autoregressive prediction setting in Section 5.2.

All experiments in this paper split the calibration set in half into equal-sized $\mathcal{D}_{cal-1}$ and $\mathcal{D}_{cal-2}$. Although the split does not significantly impact the result when calibration data is ample, performance deteriorates when there are not enough data in either one of the subsets.

**Baselines.** We compare our model with three representative works in different paradigms of deep uncertainty quantification: the Bayesian-motivated Monte Carlo dropout RNN (**MC-dropout**) by Gal & Ghahramani (2016a), the frequentist blockwise jackknife RNN (**BJRNN**) by Alaa & Van Der Schaar (2020), a conformal forecasting RNN (**CF-RNN**) by Stankevičiūtė et al. (2021), and

Table 1: Performance in synthetic and real-world datasets with target confidence $1 - \alpha = 0.9$. Methods that are *invalid* (coverage below $90\%$) are greyed out. CopulaCPTS achieves high level of calibration (coverage is close to $90\%$) while producing more efficient confidence regions.

| | | MC-dropout | BJRNN | CF-RNN | Copula | CopulaCPTS |
|---|---|---|---|---|---|---|
| Particle Sim ($\sigma = .01$) | Cov | 91.5 $\pm 2.0$ | 98.9 $\pm 0.2$ | 97.3 $\pm 1.2$ | 86.9 $\pm 1.9$ | **91.3** $\pm 1.5$ |
| | Area | 2.22 $\pm 0.05$ | 2.24 $\pm 0.59$ | 1.97 $\pm 0.4$ | 0.63 $\pm 0.07$ | **1.08** $\pm 0.14$ |
| Particle Sim ($\sigma = .05$) | Cov | 46.1 $\pm 3.7$ | 100.0 $\pm 0.0$ | 94.5 $\pm 1.5$ | 88.6 $\pm 1.7$ | **90.6** $\pm 0.6$ |
| | Area | 2.16 $\pm 0.08$ | 12.13 $\pm 0.39$ | 5.80 $\pm 0.52$ | 4.67 $\pm 0.16$ | **5.27** $\pm 1.02$ |
| Drone Sim ($\sigma = .02$) | Cov | 84.5 $\pm 10.8$ | 90.8 $\pm 2.8$ | 91.6 $\pm 9.2$ | 89.2 $\pm 1.3$ | **90.0** $\pm 0.8$ |
| | Vol | 9.64 $\pm 2.13$ | 49.57 $\pm 3.77$ | 32.18 $\pm 13.66$ | 16.92 $\pm 8.9$ | **17.12** $\pm 6.93$ |
| COVID-19 Daily Cases | Cov | 19.1 $\pm 5.1$ | 79.2 $\pm 30.8$ | 95.4 $\pm 1.9$ | 90.8 $\pm 1.4$ | **90.5** $\pm 1.6$ |
| | Area | 34.14 $\pm 0.84$ | 823.3 $\pm 529.7$ | 610.2 $\pm 96.0$ | 414.42 $\pm 5.08$ | **408.6** $\pm 65.8$ |
| Argoverse Trajectory | Cov | 27.9 $\pm 3.1$ | 92.6 $\pm 9.2$ | 98.8 $\pm 1.9$ | 89.7 $\pm 0.9$ | **90.2** $\pm 0.1$ |
| | Area | 127.6 $\pm 20.9$ | 880.8 $\pm 156.2$ | 396.9 $\pm 18.67$ | 107.2 $\pm 9.56$ | **126.8** $\pm 12.22$ |

the multi-target copula algorithm that does not have the two step calibration (**Copula**) by Messoudi et al. (2021). We use the same underlying prediction model for post-hoc uncertainty quantification methods BJRNN, CF-RNN, and CopulaCPTS. The MC-dropout RNN is of the same architecture but is trained separately, as it requires an extra dropout step during training and inference.

**Metrics.** We evaluate calibration and efficiency for each method. For calibration, we report the empirical coverage on the test set. Coverage should be as close to the desired confidence level $1 - \alpha$ as possible. Coverage is calculated as:

$$\text{Coverage}_{1-\alpha} = \mathbb{E}_{x,y \sim \mathcal{Z}} \mathbb{P}[\mathbf{y} \in \Gamma^{1-\alpha}(x)] \approx \frac{1}{|\mathcal{D}_{test}|} \sum_{x^i, y^i \in |\mathcal{D}_{test}|} \mathbb{1}(y^i \in \Gamma^{1-\alpha}(x^i)).$$

For efficiency, we report the average area (2D) or volume (3D) of the confidence regions. The measure should be as small as possible while being valid (coverage maintains above-specified confidence level). The area or volume is calculated as:

$$\text{Area}_{1-\alpha} = \mathbb{E}_{x \sim \mathcal{X}}[\|\Gamma^{1-\alpha}(x)\|] \approx \frac{1}{|\mathcal{D}_{test}|} \sum_{x^i \in |\mathcal{D}_{test}|} \|\Gamma^{1-\alpha}(x^i)\|.$$

## 5.1 SYNTHETIC DATASETS

We first test the effectiveness of our models on two synthetic spatiotemporal datasets - interacting particle systems (Kipf et al., 2018), and drone trajectory following simulated with PythonRobotics (Sakai et al., 2018). For particle simulation, we predict $\mathbf{y}_{t+1:t+h}$ where $t = 35$, $h = 25$ and $y_t \in \mathbb{R}^2$; for drone simulation $t = 60$, $h = 10$, and $y_t \in \mathbb{R}^3$. To add randomness to the tasks, we added Gaussian noise of $\sigma = .01$ and $.05$ to the dynamics of particle simulation and $\sigma = .02$ to drone dynamics. We generate 5000 samples for each dataset, and split the data by 45/45/10 for train, calibration, and test, respectively. For baselines that does not require calibration, the calibration split is used for training the model. Please see Appendix C.1 for forecaster model details.

We visualize the calibration and efficiency of the methods in Figure 3 for confidence levels $1 - \alpha = 0.5$ to $0.95$. We can see that Copula-RNN, the red lines, are more calibrated and efficient compared to other baseline methods, especially when the confidence level is high ($90\%$ and $95\%$). We can see that for harder tasks (particle $\sigma = 0.05$, and drone trajectory prediction), MC-Dropout is overconfident, whereas BJ-RNN and CF-RNN produce very large (hence inefficient) confidence regions. This behavior of CF-RNN is expected because they apply Bonferroni correction to account for joint prediction for multiple time steps, which is an upper bound of copula functions. Numerical results for confidence level $90\%$ are presented in Table 1. A qualitative comparison of confidence regions for drone simulation can be found in Figure 9 in Appendix C.4.

## 5.2 REAL WORLD DATASETS

**COVID-19.** We replicate the experiment setting of Stankevičiūtė et al. (2021) and predict new daily cases of COVID-19 in regions of the UK. The models take 100 days of data as input and forecast 50 days into the future. We used 200 time series for training, 100 for calibration, and 80 for testing.

Figure 3: Calibration (upper row) and efficiency (lower row) comparison on different $1 - \alpha$ levels for synthetic data sets. Shaded regions are $\pm\,2$ standard deviations over 3 runs. For calibration, the goal is to stay above the green dotted (validity) and coincide as closely as possible (calibration). CopulaCPTS is more calibrated across different significance levels. For efficiency, we want the metric to be small. CopulaCPTS outperforms the baselines consistently. (MC-dropout for the right two experiments produces invalid regions, so we don't consider its efficiency.)

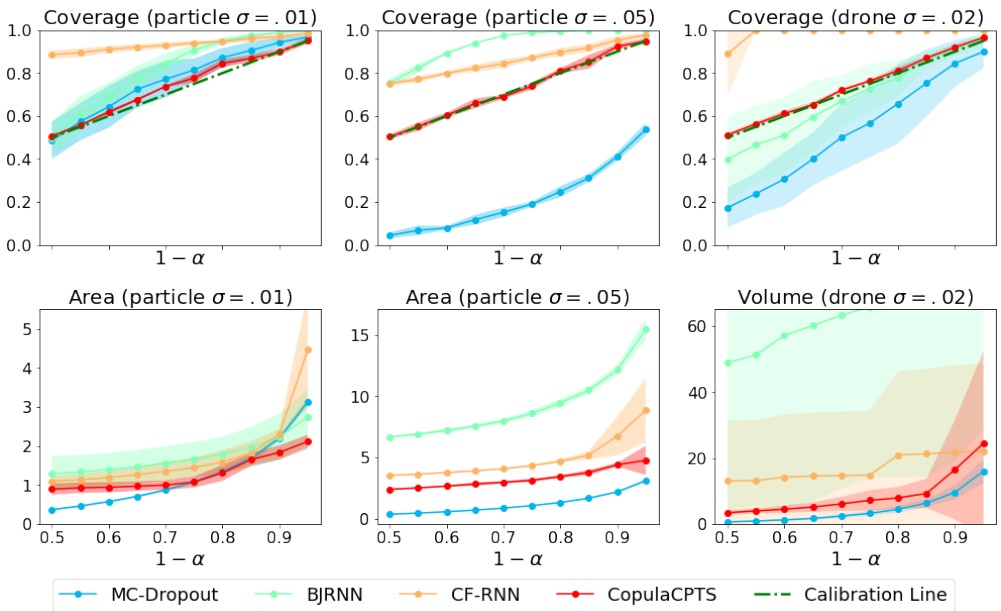

**Vehicle trajectory prediction.** The Argoverse autonomous vehicle motion forecasting dataset (Chang et al., 2019) is a widely used vehicle trajectory prediction benchmark. The task is to predict 3 second trajectories based on all vehicle motion in the past 2 seconds sampled at 10Hz. Because trajectory prediction is a challenging task, we utilize a state-of-the-art prediction algorithm LaneGCN (Liang et al., 2020) as the underlying model for both CF-RNN and Copula-RNN (details in Appendix C.1). Flexibility of underlying forecasting model is an advantage of post-hoc UQ methods such as conformal prediction. For model-dependent baselines MC-dropout and BJRNN, we have to train an RNN forecasting model from scratch for each method, which induces additional computational cost.

CopulaCPTS is both more calibrated and efficient compared to baseline models for real-world datasets (Table 1). The Covid-19 dataset demonstrates a potential failure case for our model when calibration data are scarce. Because there are only 100 calibration data, CDF and copula estimations are more stochastic depending on the dataset split, resulting in 1 case of invalidity among 3 experiment trials. Even so, CopulaCPTS shows strong performance on average by remaining valid and reducing the confidence width by 33%. For the trajectory prediction task, learning the copula results in a 40% sharper confidence region while still remaining valid for the 90% confidence interval. We visualize two samples from each dataset in Figure 3.The importance of efficiency in these scenarios is clear - the confidence regions need to be narrow enough for them to be useful for decision making. Given the same underlying prediction model, we can see that CopulaCPTS produces a much more efficient region while still remaining valid.

**Comparison of models at different horizon lengths.** CopulaCPTS is an algorithm designed to produce calibrated and efficient confidence regions for multi-step time series. When the prediction horizon is long, CopulaCPTS's advantage is more pronounced. Figure 5 shows the performance comparison over increasing time horizons on the particle dataset. See Table 3 of Appendix C for numerical results. CopulaCPTS achieves a 30% decrease in area at 20 time steps compared to CF-RNN, the best performing baseline; the decrease is above 50% at 25 time steps. This experiment shows significant improvement of using copula to model the joint distribution of future time steps.

Figure 4: Illustrations of 90% confidence regions given by CF-RNN (blue) and CopulaCPTS (orange) on two real-world datasets, COVID-19 forecast (left 2) and Argoverse (right 2 at time steps 1, 10, 20, and 30). For the Argoverse data, The red dotted lines (ego agent) and blue dotted lines (other agents) are input to the underlying prediction model and the red solid lines are the prediction output. Note that the confidence region produced CF-RNN is uninformatively large, as it covers all the lanes: these examples illustrate the importance of efficiency. Overall, CopulaCPTS is able to produce much more efficient confidence regions while maintaining valid coverage.

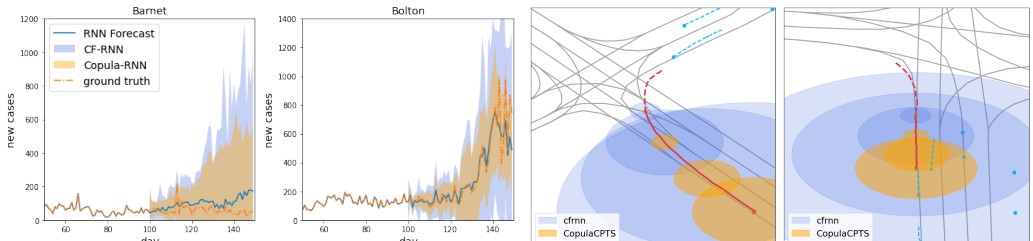

**CopulaCPTS for Autoregressive prediction.** The autoregressive extension of CopulaCPTS is illustrated in detail in Appendix B.3. To provide preliminary evidence of effectiveness, we present test results on the COVID-19 dataset. We train an RNN model with $k = 7$ and use it to autoregressively forecast the next 14 steps. Table 2 compares the performance of re-estimating the copula for each 7-step forecasts versus using a fixed copula calibrated using the first 7 steps. We also compare the model to a 14-step joint forecaster using CopulaCPTS. It is evident that daily cases of the pandemic is a non-stationary time series, where re-estimating the copula is necessary for validity.

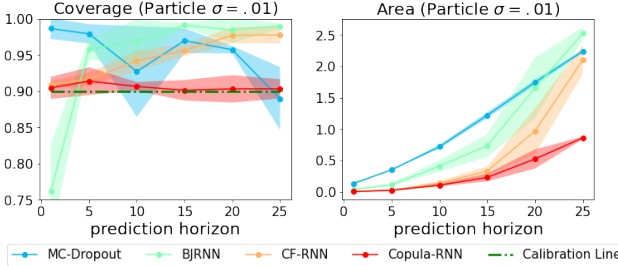

| Method | Coverage | Area |
|---|---|---|
| AR re-estimate | **90.1** | **89.4** |
| AR fixed | 88.2 | 75.9 |
| Joint | 90.7 | 102.3 |

Table 2: Performance of autoregressive (AR) CopulaCPTS. Re-estimating copula gives us valid confidence region over time and is more efficient than joint CopulaCPTS forecast.

Figure 5: CopulaCPTS remains more calibrated and efficient than baselines over increasing forecast horizons.

## 6 CONCLUSION AND DISCUSSION

In this paper, we present CopulaCPTS, a conformal prediction algorithm for multi-step time series prediction. CopulaCPTS significantly improves calibration and efficiency of multi-step conformal confidence intervals by incorporating copulas to model the joint distribution of the uncertainty at each time step. We prove that CopulaCPTS has a finite sample validity guarantee over the entire prediction horizon. Our experiments show that CopulaCPTS produces confidence regions that are (1) valid, and (2) more efficient than state-of-the-art UQ methods on all 4 benchmark datasets, and over varying prediction horizons. The improvement is especially pronounced when the data dimension is high or the prediction horizon is long, cases when other methods are prone to be highly inefficient. Hence, we argue that our method is a practical and effective way to produce useful uncertainty quantification for machine learning forecasting models.

The limitations of our algorithm are as follows. As CopulaCPTS requires two calibration steps, it is suitable only when there are abundant data for calibration. The validity proof relies on using the empirical copula, so it does not apply to other learnable copula classes. Future work includes (1) improving the autoregressive extension of CopulaCPTS, to achieve coverage over the whole horizon, and (2) developing online settings of CopulaCPTS for decision making.

ACKNOWLEDGEMENT

This work was supported in part by Army-ECASE award W911NF-23-1-0231, the U.S. Department Of Energy, Office of Science under #DE-SC0022255, IARPA HAYSTAC Program, CDC-RFA-FT-23-0069, NSF Grants #2205093, #2146343, and #2134274.

We would like to thank Bo Zhao for her helpful comments on the paper.

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

# A    PROOF OF THEOREM 4.1

**Theorem A.1** (Validity of CopulaCPTS). *The confidence region provided by CopulaCPTS (algorithm 1) is valid. i.e. $\mathbb{P}[\forall j \in \{1,\dots,k\}, y_{t+j} \in \Gamma_j^{1-\alpha}] \geq 1 - \alpha$.*

*Proof.* Define notations to be the same as in Section 4. Let $\mathcal{D} = \{z^i = (x^i, y^i)\}_{i=1}^n$ be a dataset with input $x^i \in \mathbb{R}^{t \times d}$, a time series with length $t$, and output $y^i \in \mathbb{R}^{k \times d}$ a time series with length $k$. Each data sample (of entire time series, not time step) $z^i = (x^i, y^i)$ is drawn i.i.d. from an unknown distribution $\mathcal{Z}$. This means that any other sample drawn $\mathcal{Z}$ is *exchangeable* with $\mathcal{D}$. from Dataset $\mathcal{D}$ is divided into training set $\mathcal{D}_{train}$ and two calibration sets $\mathcal{D}_{cal-1}$ and $\mathcal{D}_{cal-2}$.

We have nonconformity score function $A$ with prediction model $\hat{f}$ trained on $\mathcal{D}_{train}$. For each data point $z^i = (x^i, y^i) \in \mathcal{D}_{cal}$, we calculate the nonconformity score for each time step $j$, concatenating them into a vector $s^i$ of dimension $k$.

$$s_j^i = A(z^i)_j \stackrel{\text{e.g.}}{=} \|y_j^i - \hat{f}(x^i)_j\|, \; j = 1,\dots,k \tag{9}$$

Let $\mathcal{S}1 = \{s^i\}_{z^i \in \mathcal{D}_{cal-1}}$ be the set of nonconformity scores of data in $\mathcal{D}_{cal-1}$ and $\mathcal{S}2 = \{s^i\}_{z^i \in \mathcal{D}_{cal-2}}$ the set of nonconformity scores of data in $\mathcal{D}_{cal-2}$. Subscript $j$ will be used to index the set of specific time steps of the scores: $\mathcal{S}1_j = \{s_j^i\}_{z^i \in \mathcal{D}_{cal-1}}, \mathcal{S}2_j = \{s_j^i\}_{z^i \in \mathcal{D}_{cal-2}}$.

**CDF Estimation on $\mathcal{D}_{cal-1}$.**    We use $\mathcal{D}_{cal-1}$ to build conformal predictive distributions (CPD) (Vovk et al., 2017) for each time step's anomaly scores. The cumulative distribution function is constructed as:

$$\hat{F}_j(s_j) := \frac{1}{|\mathcal{S}1_j| + 1} \sum_{s^i \in \mathcal{S}1_j \cup \{\infty\}} \mathbb{1}_{s_j^i < s_j}, \quad \text{for } j \in \{1,\dots,k\} \tag{10}$$

**Lemma A.2** (Validity of CPD. Theorem 11 of Vovk et al. (2017) ). *Given a nonconformity score function $A : \mathcal{Z} \to \mathbb{R}$ and a data sample $z \sim \mathcal{Z}$, calculate the nonconformity score as $s = A(z)$. Then, the distribution $\hat{F}_j(\cdot)$ is valid in the sense that $\mathbb{P}_{\mathcal{Z}}[\hat{F}_j(s_j) \leq 1 - \alpha] = 1 - \alpha$, for any $0 < \alpha < 1$ .*

**Copula Calibration on $\mathcal{D}_{cal-2}$.**    Next, for every data point $\mathcal{D}_{cal-2}$, we calculate

$$\mathcal{U} = \{\mathbf{u}^i\}_{i \in \mathcal{D}_{cal-2}}, \quad \mathbf{u}^i = (u_1^i, \dots, u_k^i) = (\hat{F}_1(s_1^i), \dots, \hat{F}_k(s_k^i))$$

Each $\mathbf{u}^i$ can be seen as a *multivariate nonconformity score* for data sample $z^i$. We will now illustrate that an empirical copula on $\mathcal{U}$ is a rank statistic, and hence we can apply the proof of conformal prediction to prove a finite sample validity guarantee.

**Definition A.3** (Vector partial order). Define a partial order for $k$-dimensional vectors $\preceq$.

$$\mathbf{u} \preceq \mathbf{v} \quad \text{i.f.f.} \quad \forall j \in \{1,\dots,k\}, \; \mathbf{u}_j \leq \mathbf{v}_j \tag{11}$$

$$\text{i.e. } \mathbf{u} \preceq \mathbf{v} \iff \prod_{j=1}^k \mathbb{1}_{\mathbf{u}_j \leq \mathbf{v}_j} \tag{12}$$

Next, we define an empirical multivariate quantile function for $\mathcal{U}$, a set of $k$-dimensional vectors, based on the partial order defined in Eqn 11. [2]

$$\hat{\mathbf{Q}}(1 - \alpha, \mathcal{U}) = \arg\min_{\mathbf{u}^*} \sum_{j=1}^k \mathbf{u}_j^* \quad \text{s.t. } \left(\frac{1}{|\mathcal{U}|} \sum_{\mathbf{u} \in \mathcal{U}} \mathbb{1}_{\mathbf{u} \preceq \mathbf{u}^*}\right) \geq 1 - \alpha \tag{13}$$

The empirical copula formula in CopulaCPTS (Eqn 7 in section 4.1) is the same as the expression inside the inf function of $Q(1 - \alpha, \mathcal{U} \cup \{\infty\})$. Therefore, finding $s_1^*, \dots, s_k^*$ by Equation 8 implies:

$$\hat{\mathbf{Q}}(1 - \alpha, \mathcal{U} \cup \{\infty\}) = z$$

---

[2]There exist other definitions of multivariate quantiles, but they cannot be used in place of our definition in this proof. We have chosen this form because it connects directly to the empirical copula.

The rest of the proof follows that of Inductive Conformal Prediction (ICP) in Vovk et al. (2005).

Given a test data sample $z^{n+1} = (x_{1:t}^{n+1}, y_{1:k}^{n+1}) \sim \mathcal{Z}$, we want to prove that the confidence regions $\Gamma_1^{1-\alpha}, \ldots, \Gamma_k^{1-\alpha}$ output by CopulaCPTS satisfies:

$$\mathbb{P}[y_j \in \Gamma_j^{1-\alpha}] \geq 1 - \alpha, \quad \forall j \in \{1, \ldots, k\}$$

We first calculate

$$\mathbf{u}_j^{n+1} = \hat{F}_j(A(z^{n+1})_j) \text{ for } j \in \{1, \ldots, k\}$$

Let $\mathbf{u}^* = Q(1 - \alpha, \mathcal{U} \cup \{\infty\})$, $\mathbf{u}^* \in [0, 1]^k$. An important observation for the conformal prediction proof is that if $\mathbf{u}^* \preceq \mathbf{u}^{n+1}$, then

$$\hat{\mathbf{Q}}(1 - \alpha, \mathcal{U} \cup \{\infty\}) = \hat{\mathbf{Q}}(1 - \alpha, \mathcal{U} \cup \{\mathbf{u}^{n+1}\})$$

the quantile remains unchanged. This fact can be re-written as

$$\mathbf{u}^{n+1} \preceq \hat{\mathbf{Q}}(1 - \alpha, \mathcal{U} \cup \{\infty\}) \Longleftrightarrow \mathbf{u}^{n+1} \preceq \hat{\mathbf{Q}}(1 - \alpha, \mathcal{U} \cup \{\mathbf{u}^{n+1}\})$$

The above describes the condition where $\mathbf{u}^{n+1}$ is among the $\lceil (1 - \alpha)(n + 1) \rceil$ smallest of $\mathcal{U}$. By exchangability, the probability of $\mathbf{u}^{n+1}$'s rank among $\mathcal{U}$ is uniform. Therefore,

$$\mathbb{P}[\mathbf{u}^{n+1} \preceq \hat{\mathbf{Q}}(p, \mathcal{U} \cup \{\infty\})] = \frac{\lceil (1 - \alpha)(|\mathcal{U}| + 1) \rceil}{(|\mathcal{U}| + 1)} \geq 1 - \alpha$$

Hence we have

$$\mathbb{P}[\mathbf{u}^{n+1} \preceq \hat{\mathbf{Q}}(1 - \alpha, \mathcal{U} \cup \{\infty\})] \geq 1 - \alpha \tag{14}$$

Note again that:

- $\mathbf{u}^* = \hat{\mathbf{Q}}(1 - \alpha, \mathcal{U} \cup \{\infty\}) = (\hat{F}_1(s_1^*), \ldots, \hat{F}_t(s_k^*))$

- $\mathbf{u}^{n+1} = (\hat{F}_1(s_1^{n+1}), \ldots, \hat{F}_t(s_k^{n+1}))$

- The uncertain regions are constructed as (Algorithm 1, line 17):

$$\Gamma_j^{1-\alpha} \leftarrow \{\mathbf{y} : \|\mathbf{y} - \hat{\mathbf{y}}_j^{n+1}\| < s_j^*\} \tag{15}$$

By definition of $\preceq$, we have

$$\mathbf{u}^{n+1} \preceq \mathbf{u}^* \tag{16}$$

$$\stackrel{(11)}{\Longleftrightarrow} \forall j \in \{1, \ldots, k\}, \mathbf{u}_j^{n+1} \leq \mathbf{u}_j^* \tag{17}$$

$$\stackrel{\text{Lemma A.2}}{\Longrightarrow} \forall j \in \{1, \ldots, k\}, s_j^{n+1} \leq s_j^* \tag{18}$$

$$\stackrel{(15)}{\Longleftrightarrow} \forall j \in \{1, \ldots, k\}, \mathbf{y}_j \in \Gamma_j^{1-\alpha} \tag{19}$$

Combining Eqn 14 and Eqn 19, we have

$$\mathbb{P}[\forall j \in \{1, \ldots, k\}, \mathbf{y}_j \in \Gamma_j^{1-\alpha}] \geq \mathbb{P}[\mathbf{u}^{n+1} \preceq \hat{\mathbf{Q}}(1 - \alpha, \mathcal{U} \cup \{\infty\})] \geq 1 - \alpha \tag{20}$$

$$\square$$

## B    ADDITIONAL ALGORITHM DETAILS

### B.1    UPPER AND LOWER BOUNDS FOR COPULAS

To provide a better understanding of the properties of Copulas, consider the Frechet-Hoeffding Bounds (Theorem B.1). In fact, the Frechet-Hoeffding upper- and lower- bounds are both copulas. The lower bound is precisely the Bonferroni correction used in Stankevičiūtė et al. (2021) - therefore by estimating the copula more precisely instead of using a lower bound, we have a guaranteed efficiency improvement for the confidence region.

**Theorem B.1** (The Frechet-Hoeffding Bounds). *Consider a copula $C(u_1, \ldots, u_k)$. Then*

$$\max \left\{ 1 - k + \sum_{i=1}^{k} u_i, 0 \right\} \leq C(u_1, \ldots, u_k) \leq \min \left\{ u_1, \ldots, u_k \right\}$$

### B.2    NUMERICAL OPTIMIZATION WITH SGD FOR SEARCH

We continue to use the notation defined in Appendix A. The inverse of the predictive distributions (Equation 10) can be written as follows, similar to the empirical quantile function (Equation 3).

$$\hat{F}_j^{-1}(p) := \inf\{s_j : (\frac{1}{|\mathcal{S}1_j| + 1} \sum_{s^i \in \mathcal{S}1_j \cup \{\infty\}} \mathbb{1}_{s_j^i < s_j}) \geq p\} \tag{21}$$

We find the optimal $s_j^*$ in Equation 8 and Algorithm 1 by minimizing the following loss:

$$\mathcal{L}(s_1, \ldots, s_k) = \frac{1}{|\mathcal{D}_{cal-2}|} \sum_{i \in \mathcal{D}_{cal-2}} \prod_{j=1}^{k} \mathbb{1} \left[ \mathbf{u}_j^i < \hat{F}_j^{-1}(s_j) \right] - (1 - \alpha)$$

The indicator function is implemented using a sigmoid function whose input is multiplied by a constant for differentiability. A small amount of L2 regularization is added to the loss function to ensure the searched scores are as low as possible. We use the Adam optimizer and perform gradient descent for 500 steps to get the final result. The optimization process to find $\mathbf{s}^*$ typically takes a few seconds on CPU. For each run of our experiments, the calibration and prediction steps of CopulaCPTS combined took less than 1 minute to run on an Apple M1 CPU. Please refer to the `CP` class in the reference code for implementation details.

### B.3    COPULACPTS IN AUTO-REGRESSIVE FORECASTING

Auto-regressive forecasting is a common framework in time series forecasting. So far, we have been looking at forecasts for a predetermined number of time steps $k$. One can use a fixed-length model to forecast variable horizons $k'$ autoregressively, taking the model output as part of the input. In the conformal prediction setting, we want not only to autoregressively use the point forecasts, but also to propagate the uncertainty measurement.

If the time series and uncertainty are *stationary* (for example additive Gaussian noise), the copula remains the same for any sliding window of $k$ steps, i.e. $C(u_1, \ldots, u_k) = C(u_2, \ldots, u_{k+1})$. Therefore, after finding $(u_1^*, \ldots, u_k^*)$ such that $C(u_1^*, \ldots, u_k^*) \geq 1 - \alpha$, we can simply search for $u_{k+1}^*$ such that $C(u_2^*, \ldots, u_k^*, u_{k+1}^*) \geq 1 - \alpha$. The guarantee proven in Theorem 4.1 still holds for the new estimate. In this way, we can achieve the coverage guarantee over the entire autoregressive forecasting horizon.

On the other hand, if the time series is *non-stationary*, we need to fit copulas $C_1(u_1, \ldots, u_k)$, $C_2(u_2, \ldots, u_{k+1}), \ldots, C_{k'-k}(u_{k'-k}, \ldots, u_{k'})$, one for each autoregressive prediction, which requires a calibration set with $\geq k'$ time steps. The $k'$ step autoregressive problem is then reduced to $k' - k$ multi-step forecasting problems that can be solved by CopulaCPTS. It follows that each of the autoregressive predictions is valid. Appendix B.4 provides an example scenario where re-estimating the copula is necessary for validity.

## B.4 AUTOREGRESSIVE PREDICTION

In the context of this paper to forecast autoregressively is given input $\mathbf{x}_{1:t}$ and a $k$ step forecasting model $\hat{f}$, perform prediction

$$\hat{\mathbf{y}}_{t+1:t+k} = \hat{f}(\mathbf{x}_{1:t})$$
$$\hat{\mathbf{y}}_{t+2:t+k+1} = \hat{f}(\mathbf{x}_{2:t}, \hat{\mathbf{y}}_{t+1})$$
$$\cdots$$

until all $k'$ time steps are predicted.

We now provide a toy scenario to illustrate when re-estimating the copula is necessary and improves validity. Consider a time series of three time steps $t_0, t_1, t_2$. The two scenarios are illustrated in Figure 6. In both scenarios, the mean and variance of all time steps are 0 and 1 respectively. In scenario (a), $t_0 = t_1$ and hence their covariance is 1. The copula estimated on $t_0$ and $t_1$ is $C_{0:1}(F(t_0), F(t_1)) = F(t_0) = F(t_1)$. This copula will significantly underestimate the confidence region of $t_2$ where its covariance with $t_1$ is $-1$. In fact the coverage of $C_{0:1}(F_1(t_1), F_2(t_2)) = 0.74$. On the other hand, (b) illustrates a scenario where the copula for any 2 consecutive time series remains the same $C_0 = C_1$. In this case, applying $C_0$ directly to forecast $C_1$ achieves precisely 90% coverage.

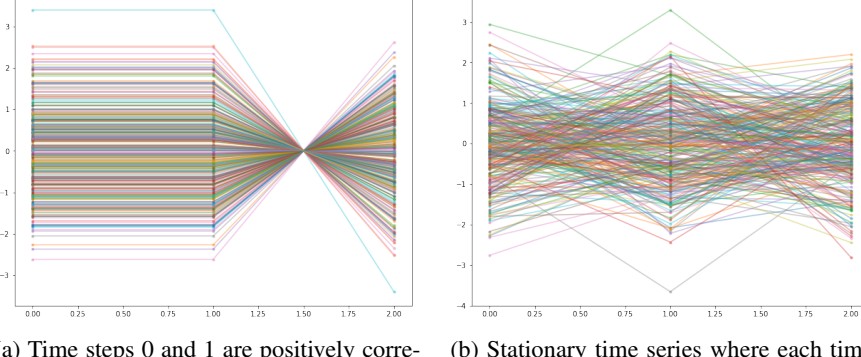

(a) Time steps 0 and 1 are positively correlated while 1 and 2 are negatively correlated

(b) Stationary time series where each time steps are uncorrelated

Figure 6: Two scenarios to illustrate the autoregressive case

## C EXPERIMENT DETAILS AND ADDITIONAL RESULTS

### C.1 UNDERLYING FORECASTING MODELS

**Particle Dataset.** The underlying forecasting model for the particle experiments is an 1-layer LSTM network with embedding size $= 24$. The hidden state is then passed through a linear network to forecast the timesteps concurrently (output has dimension $k \times d_y$). We train the model for 150 epochs with batch size 128. Hyperparameters of the network are selected through a model search by performance on a 5-fold cross-validation split of the dataset. The architecture and hyperparameters are shared for all baselines and CopulaCPTS in Table 1.

**Drone.** For the drone trajectory forecasting task, we use the same LSTM forecasting network as the particle dataset, but with its hidden size increased to 128. We train the model for 500 epochs with batch size 128. The same architecture and hyperparameters are shared for all baselines and CopulaCPTS reported in Table 1.

**Covid-19.** The COVID-19 dataset is downloaded directly from the official UK government website https://coronavirus.data.gov.uk/details/download by selecting *region* for area

type and *newCasesByPublishDate* for metric. There are in total 380 regions and over 500 days of data, depending on when it is downloaded. We selected 150-day time series from the collection to construct our dataset.

The base forecasting model for Covid-19 dataset is the same as the model for synthetic datasets, with hidden size = 128, and were trained for 150 epochs with batch size 128. The same architecture and hyperparameters are shared for all baselines and CopulaCPTS reported in Table 1.

**Argoverse.** As highlighted in the main text, we utilize a state-of-the-art prediction algorithm LaneGCN (Liang et al., 2020) as the underlying forecaster model for CF-RNN and Copula-RNN. We refer the readers to their paper and code base for model details. The architecture of the RNN network used for MC-Dropout and BJRNN is an Encoder-Decoder network. Both the encode and decoder contain a LSTM layer with encoding size 8 and hidden size 16. We chose this architecture because the is part of the official Argoverse baselines (https://github.com/jagjeet-singh/argoverse-forecasting) and demonstrates competitive performance.

## C.2 CALIBRATION AND EFFICIENCY CHART FOR COVID-19

Figure 7 shows a comparison of calibration and efficiency for the daily new COVID 19 cases forecasting.

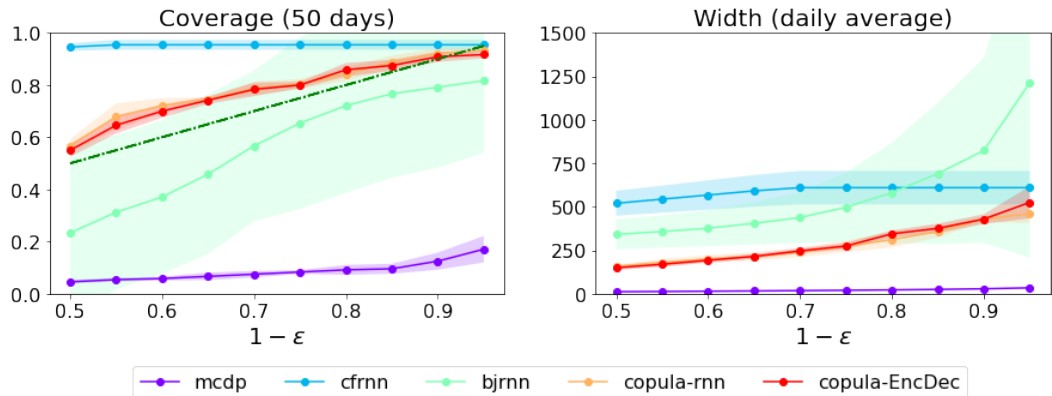

Figure 7: Calibration and efficiency comparison on different $\epsilon$ level for COVID-19 Daily Forecasts. The copula methods (orange and red lines) are more calibrated (coinciding with the green dotted line) and sharp (low width) compared to baselines.

To see if the daily fluctuation due to testing behavior disrupts other methods, we also ran the same experiment on weekly aggregated new cases forecast. We take 14 weeks of data as input and output forecasts for the next 6 weeks. The results are illustrated in Figure 8. The weekly forecasting scenario gives us similar insights as the daily forecasts.

## C.3 ARGOVERSE

The Argoverse autonomous vehicle dataset contains 205,942 samples, consisting of diverse driving scenarios from Miami and Pittsburgh. The data can be downloaded from the official Argoverse dataset website. We split 90/10 into a training set and validation set of size 185,348 and 20,594 respectively. The official validation set of size 39,472 is used for testing and reporting performance. We preprocess the scenes to filter out incomplete trajectories and cap the number of vehicles modeled to 60. If there are less than 60 cars in the scenario, we insert dummy cars into them to achieve consistent car numbers. For map information, we only include center lanes with lane directions as features. Similar to vehicles, we introduce dummy lane nodes into each scene to make lane numbers consistently equal to 650.

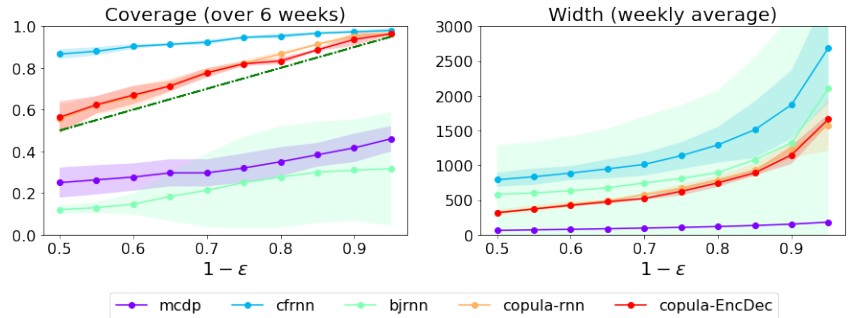

Figure 8: Covid Weekly Forecasts

## C.4  ADDITIONAL EXPERIMENT RESULTS

We present in Figures 8 and 9 some qualitative results for uncertainty estimation.

To test how the effects of copulaCPTS compare with baseline on other base forecasters, we also include an encoder-decoder architecture with the same embedding size as the RNN models introduced in Appendix C.1 for each dataset. The results are presented in Table 3. We omit these results in the main text because we found that they did not bring significant improvement to time series forecasting UQ.

Table 4 compares model performance compared across different prediction horizons. We show that the advantage of our method is more pronounced for longer horizon forecasts.

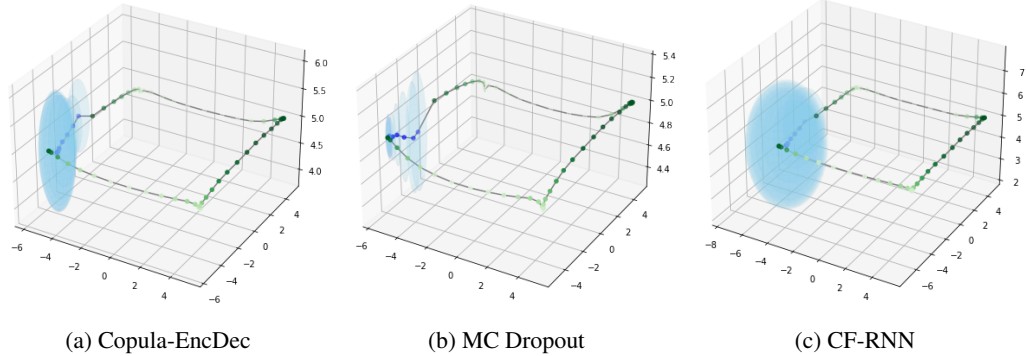

(a) Copula-EncDec          (b) MC Dropout          (c) CF-RNN

Figure 9: 99% Confidence region produced by three methods for the drone dataset. Copula methods (a) produce a more consistent, expanding cone of uncertainty compared to MC-Dropout (b) sharper one compared to CF-RNN (c).

## C.5  STUDY ON $\alpha_j$ SEARCH

Figure 11 shows the $\alpha_j$ values for each $1 - \alpha_j = \hat{F}_j(s_j^*)$ used in Copula CPTS as outlined in line 15 of Algorithm 1. We present $\alpha_j$ values searched using two methods of searching, with dichotomy search for a constant $\alpha$ value for the horizon as in Messoudi et al. (2021), and by stochastic gradient descent as outlined in section 4.2.

The $\alpha_j$ values are an indicator of how interrelated the uncertainty between each time step are: Bonferroni Correction used in Stankevičiūtė et al. (2021) (grey dotted line in Figure 11) assumes that the time steps are independent, with CopulaCPTS we have lower $1 - \alpha_j$ levels while having valid coverage (blue and orange lines in Figure 11). This shows that the uncertainty of the time steps is not independent, and we are able to utilize this dependency to shrink the confidence region while still maintaining the coverage guarantee.

| Particle Simulation ($\sigma = .01$) | | | | |
|---|---|---|---|---|
| | Coverage (90%) | Area (90%) | Coverage (99%) | Area (99%) |
| MC-dropout | 691.5 ± 2.0 | 2.22 ± 0.05 | 95.2 ± 1.4 | 3.16 ± 0.08 |
| BJRNN | 98.9 ± 0.2 | 2.24 ± 0.59 | 99.6 ± 0.3 | 2.75 ± 0.71 |
| CF-RNN | 97.1 ± 0.8 | 1.2 ± 0.21 | 99.3 ± 0.6 | 3.16 ± 0.86 |
| CF-EncDec | 97.3 ± 1.2 | 1.97 ± 0.4 | 98.9 ± 0.6 | 2.75 ± 0.42 |
| Copula-vanilla | 86.9 ± 1.9 | 0.63 ± 0.07 | 91.9 ± 1.8 | 0.76 ± 0.12 |
| Copula-RNN | 91.3 ± 1.5 | **1.08** ± 0.14 | 99.4 ± 0.3 | 2.23 ± 0.19 |
| Copula-EncDec | 90.8 ± 2.5 | 1.19 ± 0.08 | 99.3 ± 0.5 | 2.16 ± 0.23 |

| Particle Simulation ($\sigma = .05$) | | | | |
|---|---|---|---|---|
| | Coverage (90%) | Area (90%) | Coverage (99%) | Area (99%) |
| MC-dropout | 16.1 ± 4.3 | 0.79 ± 0.02 | 33.9 ± 5.1 | 2.12 ± 0.03 |
| BJRNN | 100.0 ± 0.0 | 12.13 ± 0.39 | 100.0 ± 0.0 | 15.43 ± 0.85 |
| CF-RNN | 94.5 ± 1.5 | 5.79 ± 0.51 | 99.8 ± 2.2 | 19.21 ± 8.19 |
| Copula-vanilla | 88.5 ± 1.7 | 4.37 ± 0.16 | 91.7 ± 1.6 | 4.8 ± 0.18 |
| Copula-RNN | **90.3** ± 0.7 | 4.50 ± 0.07 | **99.1** ± 0.8 | 12.82 ± 3.98 |
| Copula-EncDec | 91.4 ± 1.1 | **4.40** ± 0.15 | 98.7 ± 0.1 | **9.31** ± 1.97 |

| Drone Simulation ($\sigma = .02$) | | | | |
|---|---|---|---|---|
| | Coverage (90%) | Area (90%) | Coverage (99%) | Area (99%) |
| MC-dropout | 84.5 ± 10.8 | 9.64 ± 2.13 | 90.0 ± 7.8 | 16.02 ± 3.62 |
| BJRNN | 90.8 ± 2.8 | 49.57 ± 3.77 | 100.0 ± 4.0 | 65.77 ± 4.56 |
| CF-RNN | 91.6 ± 9.2 | 32.18 ± 13.66 | 100.0 ± 0.0 | 36.79 ± 14.03 |
| CF-EncDec | 100.0 ± 0.0 | 21.83 ± 26.29 | 100.0 ± 0.0 | 25.03 ± 12.53 |
| Copula-vanilla | 89.5 ± 1.3 | 54.67 ± 28.9 | 94.5 ± 0.5 | 68.9 ± 33.42 |
| Copula-RNN | **90.0** ± 1.5 | **16.52** ± 15.08 | **98.5** ± 0.5 | **21.48** ± 8.91 |

| COVID-19 Daily Cases Dataset | | | | |
|---|---|---|---|---|
| | Coverage (90%) | Area (90%) | Coverage (99%) | Area (99%) |
| MC-dropout | 19.1 ± 5.1 | 34.14 ± 0.84 | 100.0 ± 0.0 | 1106.57 ± 25.41 |
| BJRNN | 79.2 ± 30.8 | 823.3 ± 529.7 | 85.7 ± 27.5 | 149187. ± 51044. |
| CF-RNN | 95.4 ± 1.9 | 610.2 ± 96.0 | 100.0 ± 0.0 | 121435. ± 26495. |
| CF-EncDec | 91.7 ± 1.4 | 570.3 ± 22.1 | 100.0 ± 0.0 | 108130. ± 10889. |
| Copula-vanilla | 90.8 ± 1.4 | 414.42 ± 5.08 | 91.2 ± 1.3 | 41346. ± 59.0 |
| Copula-RNN | 92.1 ± 1.0 | **429.0** ± 15.1 | 100.0 ± 0.0 | 88962. ± 9643. |
| Copula-EncDec | **90.8** ± 0.3 | 429.4 ± 27.9 | 100.0 ± 0.0 | **60852.** ± 12263. |

| Argoverse Trajectory Prediction Dataset | | | | |
|---|---|---|---|---|
| | Coverage (90%) | Area (90%) | Coverage (99%) | Area (99%) |
| MC-dropout | 27.9 ± 3.1 | 127.6 ± 20.9 | 31.5 ± 3.9 | 242.1 ± 54.0 |
| BJRNN | 92.6 ± 9.2 | 880.8 ± 156.2 | 100.0 ± 0.0 | 3402.8 ± 268. |
| CF-LaneGCN | 98.8 ± 1.9 | 396.9 ± 18.67 | 100. ± 0.2 | 607.2 ± 8.67 |
| Copula-vanilla | 89.7 ± 0.9 | 107.2 ± 9.56 | 96.5 ± 2.3 | 289.0 ± 38.1 |
| Copula-LaneGCN | **90.4** ± 0.3 | **126.8** ± 12.22 | **99.1** ± 0.4 | **324.1** ± 42.22 |

Table 3: Additional results. Copula methods achieve a high level of calibration while producing sharper prediction regions. The sharpness gain is even more pronounced at higher confidence levels (99%), where we want the prediction region to be useful while remaining valid.

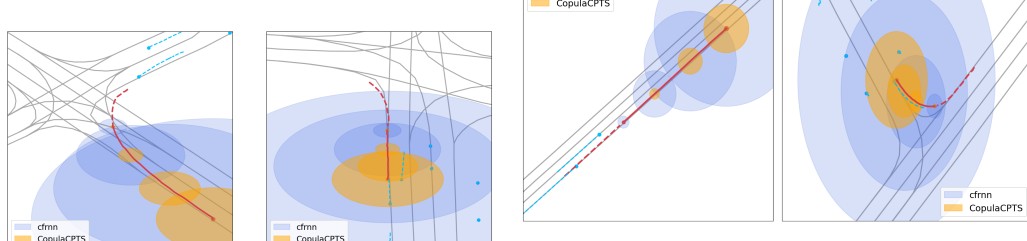

Figure 10: Illustrations for confidence regions given by CF-RNN (blue) and CopulaCPTS (orange) at time steps 0, 10, 20, and 30. Note that in order to achieve 90% coverage, the regions are larger than needed, especially in straight-lane cases like the middle two. Using copulas to couple together time steps results in a much smaller region while achieving similarly good coverage.

| Method | 1 Step | | 5 Steps | | 15 Steps | |
|---|---|---|---|---|---|---|
| | Coverage | Area | Coverage | Area | Coverage | Area |
| MC-Dropout | 97.8 $\pm$ 2.0 | 0.4 $\pm$ 0.04 | 88.0 $\pm$ 7.0 | 0.69 $\pm$ 0.25 | 52.3 $\pm$ 1.4 | 0.94 $\pm$ 0.2 |
| BJRNN | 45.3 $\pm$ 39.4 | 0.27 $\pm$ 0.18 | 97.7 $\pm$ 2.1 | 2.69 $\pm$ 1.79 | 95.5 $\pm$ 2.8 | 19.99 $\pm$ 4.83 |
| CF-RNN | 100.0 $\pm$ 0.0 | **0.01** $\pm$ 0.01 | 77.8 $\pm$ 19.2 | 0.8 $\pm$ 0.64 | 66.7 $\pm$ 0.0 | 18.82 $\pm$ 3.73 |
| CF-EncDec | 89.9 $\pm$ 19.2 | **0.01** $\pm$ 0.01 | 100.0 $\pm$ 0.0 | 0.75 $\pm$ 0.99 | 88.9 $\pm$ 19.2 | 13.07 $\pm$ 16.1 |
| Copula-RNN | 90.1 $\pm$ 0.2 | **0.01** $\pm$ 0.01 | 89.8 $\pm$ 0.6 | **0.54** $\pm$ 0.45 | **90.1** $\pm$ 1.2 | 8.25 $\pm$ 3.44 |
| Copula-EncDec | **90.0** $\pm$ 0.3 | **0.01** $\pm$ 0.0 | **90.3** $\pm$ 0.6 | 0.67 $\pm$ 1.01 | 90.5 $\pm$ 0.5 | **7.13** $\pm$ 9.5 |

Table 4: Performance comparison across different horizons at 90% confidence level on the drone simulation dataset. The improvement on efficiency is more pronounced when the horizon is longer.

Table 5 shows that there are no significant differences between coverage and area performance for the two search methods within the scope of datasets we study in this paper. However, we want to highlight that SGD search is $O(n)$ complexity to optimization steps, regardless of the prediction horizon. SGD also allows for varying $\alpha_j$ which might be useful in some settings, for example capturing uncertainty spikes for some time steps as seen in the COVID-19 dataset of Figure 11. Dichotomy search, on the other hand, is $O(nlog(n))$ complexity to the search space depends on granularity, and will be $O(knlog(kn))$ if we want to search for varying $\alpha_j$.

| Dataset | Coverage (90%) | | Area | |
|---|---|---|---|---|
| | Fixed $\alpha_j$ | Varying $\alpha_j$ | Fixed $\alpha_j$ | Varying $\alpha_j$ |
| Particle ($\sigma = .01$) | 91.7 $\pm$ 1.9 | 91.5 $\pm$ 2.1 | 1.13 $\pm$ 0.45 | 1.06 $\pm$ 0.36 |
| Particle ($\sigma = .05$) | 92.1 $\pm$ 1.3 | 90.3 $\pm$ 0.7 | 4.89 $\pm$ 0.05 | 4.50 $\pm$ 0.07 |
| Drone | 90.3 $\pm$ 0.5 | 90.0 $\pm$ 1.5 | 15.92 $\pm$ 1.98 | 16.52 $\pm$ 7.08 |
| Covid-19 | 92.9 $\pm$ 0.1 | 92.1 $\pm$ 1.0 | 498.44 $\pm$ 6.36 | 429.0 $\pm$ 15.1 |
| Argoverse | 90.2 $\pm$ 0.1 | 90.4 $\pm$ 0.3 | 117.1 $\pm$ 7.3 | 126.8 $\pm$ 12.2 |

Table 5: Coverage and area comparison between stochastic search for fixed $\alpha_j$ and SGD for Varying $\alpha_j$. We do not see a significant difference between the performance of the two.

### C.6 COMPARISON TO ADDITIONAL BASELINES

We include a comparison to two additional simple UQ baselines on the particle simulation dataset.

**L2-Conformal.** L2-Conformal uses the same underlying RNN forecaster as CF-RNN and Copula RNN. We use the nonconformity score of the vector norm of all timesteps concatenated together $\|\hat{\mathbf{y}}_{t+1:t+k} - \mathbf{y}_{t+1:t+k}\|$ to perform ICP. As there are no analytic way to represent a $k \times d_y$-dimensional

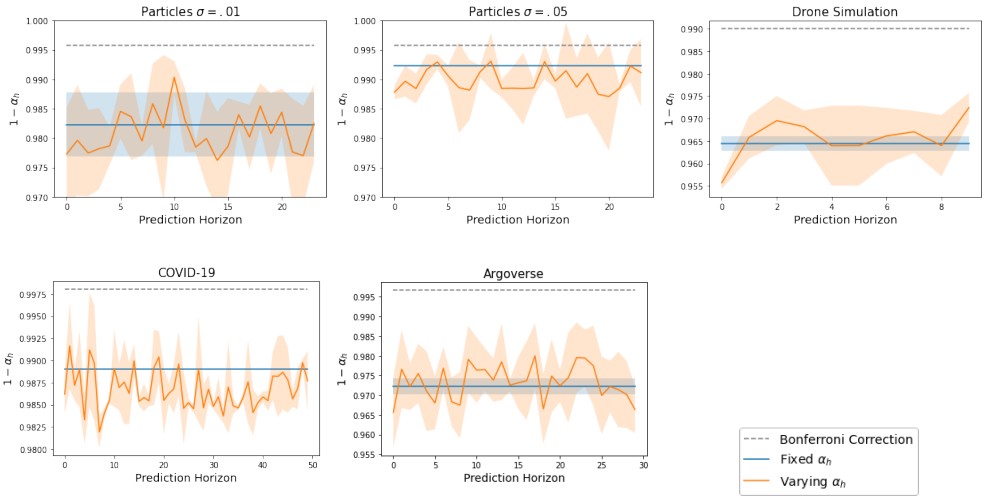

Figure 11: Comparison between dichotomy search for fixed $\alpha_j$ values (blue) and stochastic gradient search for varying $\alpha_j$ (blue) through timesteps. Shaded regions are the standard deviation of the values over 3 runs.

uncertainty region on 2-D space, we calculate the area and plot the region for L2 Conformal baseline with the maximum deviation at each timestep such that the vector norm still stays within range.

**Direct Gaussian.** Direct Gaussian uses the same model architecture and training hyperparameters, with the addition of a linear layer that outputs the variance for each timestep, and is optimized using negative log loss, a proper scoring rule for probabilistic forecasting. We obtain the area by analytically calculating the 90% confidence interval for each variable.

Results in Table 6 show that L2-conformal produces inefficient confidence area, and directly outputting variance under-covers test data. These results align with previous findings and motivate our method, which is both more calibrated and sharper compared to these baselines. We show a visualization in Figure 12 to illustrate the different properties of the methods qualitatively.

| | Particle ($\sigma = .01$) | | Particle ($\sigma = .05$) | |
|---|---|---|---|---|
| Method | Coverage (90%) | Area $\downarrow$ | Coverage (90%) | Area $\downarrow$ |
| L2-Conformal | $88.5 \pm 0.4$ | $7.21 \pm 0.35$ | $89.7 \pm 0.6$ | $7.21 \pm 0.35$ |
| Direct Gaussian | $11.9 \pm 0.09$ | $0.07 \pm 0.31$ | $0.0 \pm 0.0$ | $0.08 \pm 0.02$ |
| CF-RNN | $97.0 \pm 2.3$ | $3.13 \pm 3.24$ | $97.0 \pm 2.3$ | $5.79 \pm 0.51$ |
| CopulaCPTS | $\mathbf{91.3} \pm 2.1$ | $\mathbf{1.08} \pm 0.36$ | $\mathbf{90.3} \pm 0.7$ | $\mathbf{4.50} \pm 0.07$ |

Table 6: Comparison with two additional baselines on the particle dataset.

**Ellipsoidal conformal inference for Multi-Target Regression** We also compare CopulaCPTS to a newer work, Ellipsoidal CP (Messoudi et al., 2022). The result is presented in Table 7. This method models the uncertainty region of multi-target outputs as a high-dimensional ellipsoid, by estimating a covariance matrix on calibration data. We apply EllipsoidalCP on our data by flattening the time and space dimensions, so the particle simulation, for example, is treated as a multi-target prediction of dim $= 50 = 25$ (time steps) $\times 2$ (dims) . We see that the results are comparable in our experiment. When the correlation is more pronounced such as in the covid experiment, EllipsoidalCP can capture the correlation better than CopulaCPTS resulting in improved efficiency. On the other hand, the flexibility of our method allows us to achieve better efficiency than that of EllipsoidalCP. A notable concern for using EllipsoidalCP is that for higher output dimensions, the determinant of the covariance matrix can be extremely large (up to $10^{50}$ in our experiments) and can result in numerical instabilities.

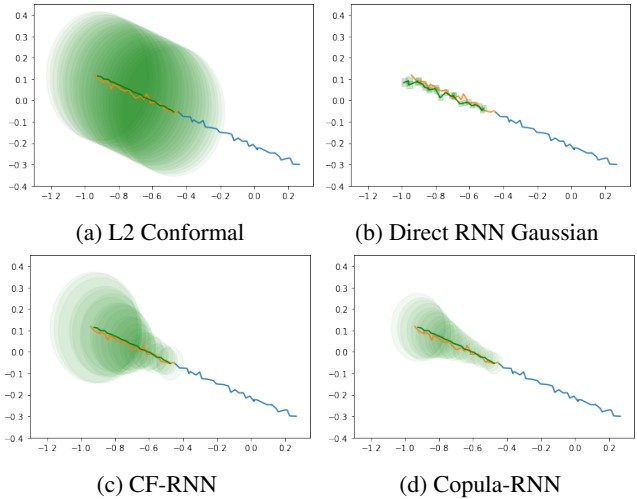

Figure 12: Visualization of on a sample from the Particle dataset's test set.

Table 7: Performance comparison with EllipsoidalCP in synthetic and real-world datasets with target confidence $1 - \alpha = 0.9$.

|  |  | EllipsoidalCP | CopulaCPTS |
|---|---|---|---|
| Particle Sim ($\sigma = .01$) | cov | **90.1** ± 0.9 | 91.3 ± 1.5 |
|  | area | **0.84** ± .005 | 1.08 ± 0.14 |
| Particle Sim ($\sigma = .05$) | cov | 90.8 ± 0.4 | **90.6** ± 0.6 |
|  | area | 8.76 ± 0.41 | **5.27** ± 1.02 |
| Drone Sim ($\sigma = .02$) | cov | 90.5 ± 0.2 | **90.0** ± 0.8 |
|  | area | 28.3 ± 3.1 | **17.12** ± 6.93 |
| COVID-19 Daily Cases | cov | 93.3 ± 1.5 | **90.5** ± 1.6 |
|  | area | **231.5** ± 22.4 | 408.6 ± 65.8 |
| Argoverse Trajectory | cov | 90.3 ± 0.1 | **90.2** ± 0.1 |
|  | area | 144.8 ± 8.1 | **126.8** ± 12.2 |

