# OpenReview forum: "Copula Conformal prediction for multi-step time series prediction"
_ICLR.cc/2024/Conference — ICLR 2024 poster_

### Official Review · Reviewer_ruzn · 2023-10-23

**Soundness:** 3 good
**Presentation:** 3 good
**Contribution:** 2 fair
**Rating:** 5
**Confidence:** 4

**Summary:**

This work considers conformal prediction for multi-step time-series prediction using copula, demonstrating improved performance than existing baselines.

**Strengths:**

* The idea of multi-step time-series forecasting using copula CP is novel.
* Strong empirical performance on various tasks.

**Weaknesses:**

1. Theoretical analyses: the i.i.d. assumption is imposed on $z^i=(x^i_{1:t}, y^i_{1:t})$ for $i=1,...,n$. However, this eliminates the dependency of data over time, which is typical and expected for time-series. Hence, this seems very restrictive for the purpose of theoretical analyses, and it is important to discuss what this assumption actually imposes on the data collection process.

2. The simulation performance of CopulaCPTS seems very similar to Copula [Messoudi et al. 2021], which notably was not developed for this setting. I think this goes back to the assumption on $z^i$ being i.i.d., where the use of CopulaCPTS is not essential under the absence of temporal dependency.

3. Performance of Copula on real-data examples is not reported.

4. Given that Copula does not seem to under-perform much, I think additional comparison is needed. For example, the locally ellipsoid CP in [Ellipsoidal conformal inference for Multi-Target Regression](https://copa-conference.com/papers/COPA2022_paper_7.pdf) by [Messoudi et al. 2022]

**Questions:**

No additional questions are raised.

---

> ### Author Response · Authors · 2023-11-18
>
> We thank the reviewer for the detailed review.
>
> Our response to the weaknesses are:
>
> > 1. Theoretical analyses: the i.i.d. assumption is imposed onHowever, this eliminates the dependency of data over time, which is typical and expected for time-series.
>
> This is a major misunderstanding. The superscript i is the index of the data sample in the dataset; we are assuming each time series sample  is independent, not the timesteps (denoted by underscript). Please refer to our notation in the second paragraph of section 4. Therefore, our method maintains the temporal dependency. Our figure 1 helps illustrate our setting of dependent time steps and independent data samples of whole trajectories. We understand that this could be confusing, so we added a parenthesis in the proof. Hope this helps to clear things out.
>
> --
>
> > 2. The simulation performance of CopulaCPTS seems very similar to Copula [Messoudi et al. 2021], which notably was not developed for this setting.
>
> We respectfully disagree and argue that the difference in both the algorithm and empirical  performance is significant. Note that in all of the Copula [Messoudi et al.] results in table 1, the coverage is invalid (below 90%), whereas all of the CopulaCPTS coverages are valid. Because the main appeal of conformal methods is the guaranteed validity, our algorithm brings significant theoretical and practical improvement.
>
> Messoudi et al.’s work also assumes the multitarget data samples (equivalent to our $z^i$) are independent. You are right in that one can treat multi-step prediction as a multitarget problem and use [Messoudi et al.] directly, but I hope our strong experiment results in table 1 show the reason why we need to use CopulaCPTS instead of [Messoudi et al].
>
> --
>
> > 3. Performance of Copula on real-data examples is not reported.
>
> Please see the last two rows of table 1 for quantitative results, figure 4 for qualitative examples, and figure 7,8, and 9 in the appendix for visualizations on real data.
>
> --
>
>
> > 4. [Messoudi et al. 2021] does not underperform much; Need more baselines.
>
> In our response to weakness #2, we have shown that CopulaCPTS significantly improves upon  [Messoudi et al. 2021] for our setting. We have added the Locally ellipsoidal CP as a baseline in table 7 in appendix C.7. The two methods achieve comparable results.

---

### Official Review · Reviewer_bjQ8 · 2023-10-31

**Soundness:** 3 good
**Presentation:** 3 good
**Contribution:** 3 good
**Rating:** 8
**Confidence:** 4

**Summary:**

The paper studies a special setting of multi-step time series forecasting with the focus on predicting confidence intervals when a traning dataset of similar time series is available.  The authors combine conformal prediction with copula modelling in a two-step algorithm that uses a part of the training dataset for calibration of confidence intervals and can be based on any other time series forecasting model. They prove the validity of the introduced algorithm and evaluate its performance on two synthetic and two real-world datasets.

**Strengths:**

## Originality
The paper presents a novel application of copula based conformal prediction explored in multi-target regression to time series setting
## Quality
* The presented method is both theoretically grounded and practically applicable
* Authors discuss the limitations of the algorithm
* Authors showcase the performance of their algorithm from different angles including exemplar visualizations
## Clarity
The paper is well-written and the presentation is very clear
## Significance
The paper is a continuation of an existing line of work on copula based conformal prediction into time series domain. Due to method limitations, in particular the requirements for vast calibration set size for non-parametric modelling, I would expect the paper to be of limited significance.

**Weaknesses:**

The paper doesn't have major weaknesses. The main downside of the presented method is its relience on large calibration dataset, which is a luxury in practical problems.

**Questions:**

* Equation (4) presents validity in terms of marginal distributions, while the prior work, e.g. Stankeviciute et al. (2021) uses a joint distribution. The proof of your theorem also uses the definition based on the joint distribution. Which one is correct?
* In experiments, did you use the same training dataset for all methods or did you adapt it depending on whether the method need a calibration set? I assume you used the same one for all baselines, but it seems fair to give methods that don't rely on calibration more data.

---

> ### Author Response · Authors · 2023-11-18
>
> We thank the reviewer for your thoughtful and generous review.
>
> To answer your questions:
>
> > Equation (4) different from prior work.
>
> Apologies for this typo, the $\forall j$ should be inside the probability, as we are certifying the joint coverage of all timesteps. We have updated our draft for the expression both in equation (4) and theorem 4.1 in the new version.
>
>
> --
>
>
> >  In experiments, did you use the same training dataset for all methods or did you adapt it depending on whether the method need a calibration set?
>
> Thank you for helping to clarify this! For baselines that do not require calibration, the calibration split is used for training the model. We have added this sentence in section 5.1.

---

> > ### Comment · Reviewer_bjQ8 · 2023-11-22
> >
> > After reading other reviews and author responses to them, I am inclined to maintain my score.

---

### Official Review · Reviewer_XEiW · 2023-10-31

**Soundness:** 3 good
**Presentation:** 3 good
**Contribution:** 3 good
**Rating:** 6
**Confidence:** 3

**Summary:**

This paper introduces a method called CopulaCPTS which allows to perform conformal prediction to any multivariate multi-step forecaster, with statistical guarantees of validity. Based on the notion of copula to model the dependency between forecasted time steps, the authors prove that CopulaCPTS has finite sample validity guarantee. On both synthetic and real multivariate time series, they show that CopulaCPTS produces more efficient confidence intervals than existing techniques.

**Strengths:**

-- The analysis of time series is a very interesting problem in conformal prediction.

-- The experience shows that the method performs quite well.

-- The paper is clear and easy to follow.

**Weaknesses:**

-- It seems that the algorithm needs a large amount of data as the calibration dataset is split a second time.

-- In the experiment, computational times are not given.

-- The standard deviations of the experiments are made with only 3 runs of the algorithm. Maybe the algorithm is very time-consuming (this should be more discuss).


Minor:

-- 'the the green'

-- Lemma ?? (eq 18)

-- In the right-hand side of equation 12, there should be a $v$ and not a $u^*$ (?).

**Questions:**

-- The paper uses a particular definition for the multivariate empirical quantile function (Eq. 13). Is this the only possible definition? If not, why use this one and not another?

-- In Lemma A.2, how is it possible to have an equality? (Do scores not need to be continuous?).

-- Isn't the definition of "exchangeability" in the paper rather a consequence of the "true" definition?

-- Is the "$\forall j$" in the probabilty or outside ? In the proof, Eq. (20), this is inside but in Theorem 4.1, this is outside.

-- In the experiments, the score is chosen to be an L2 norm (see, for example, step 9 of Algorithm 1). What are the implications of this choice on the results? For example, are the results very different if we use another norm?

---

> ### Author Response · Authors · 2023-11-18
> **Response**
>
> We thank the reviewer for reading our paper carefully and writing a helpful review. Appreciate it!
>
> In response to the weaknesses (W) and questions (Q):
>
> > W2: In the experiment, computational times are not given. The standard deviations of the experiments are made with only 3 runs of the algorithm. Maybe the algorithm is very time-consuming (this should be more discuss).
>
> Thank you for pointing out this potential source of confusion! CopulaCPTS is not time-consuming to run. It does not grow exponentially with prediction horizon, because we search for $\mathbf{u}^*$ by SGD (appendix B.2). We ran the experiment for 3 runs because it is common practice in the machine learning literature, and that our runtime bottleneck was the baseline BJRNN. We have added this discussion paragraph in the appendix B.2 on SGD:
>
> " … The optimization process to find $\mathbf{s}^*$ typically takes a few seconds on CPU. For each run of our experiments, the calibration and prediction steps of CopulaCPTS combined took less than 1 minute to run on an Apple M1 CPU. "
>
>
>
> --
>
>
>
> > W3: typos
>
> We have fixed the typos in our draft. Thank you for catching them!
>
>
>
> --
>
>
> > Q1: Particular definition for the multivariate empirical quantile function (Eq. 13)
>
> We constructed this vector empirical quantile function for the algorithm. Since comparisons between vectors are ambiguous, one cannot directly use a canonical empirical quantile function such as equation 10. Hence, we introduce the partial order and define a specific quantile function based on the partial order.
>
>
> --
>
>
> > Q2: In Lemma A.2, how is it possible to have an equality? (Do scores not need to be continuous?).
>
> The scores are continuous. The probability in lemma A.2 is a cumulative probability of the value of $\hat{F}_j(s_j)$, so the scores can be continuous and the equality can hold. Please let me know if this answers your question.
>
>
>
> --
>
>
> > Q3: Isn't the definition of "exchangeability" in the paper rather a consequence of the "true" definition?
>
> This definition of exchangeability is commonly used in papers on conformal prediction beginning from [Vovk et al. 2005]. We use this formulation as the “exchangeability assumption” in our paper for clarity and brevity.
>
>
> --
>
>
> > Q4: is the "∀j" in the probability or outside ? In the proof, Eq. (20), this is inside but in Theorem 4.1, this is outside.
>
> The $ \forall j $ should be inside the probability as in Eq 20, as we are trying to certify the joint probability here. Thank you for catching this important typo, we have updated Theorem 4.1 in the new version.
>
>
> --
>
>
> > Q5: In the experiments, the score is chosen to be an L2 norm (see, for example, step 9 of Algorithm 1). What are the implications of this choice on the results? For example, are the results very different if we use another norm?
>
> The beauty of conformal prediction is that the validity guarantee is independent of the choice of nonconformity scores. Our proof is also general with regard to which score is used. Because we model the copula directly on the empirical CDF and hence the rank of nonconformity scores, our efficiency argument is also generalizable to any score choice. There are works exploring domain-specific nonconformity scores to improve efficiency, but we regard that as an orthogonal line of work to ours.

---

> > ### Comment · Reviewer_XEiW · 2023-11-21
> >
> > I would like to thank the authors for their detailed response.
> >
> > 1\ Q2: In Lemma A.2, how is it possible to have an equality? (Do scores not need to be continuous?).
> >
> > It seems to me that it is never clearly stated in the paper that the scores are continuous (the choice of the l2 norm as the score function is presented as an example). I think this should be indicated in the theoretical statements (e.g. in lemma A.2).
> >
> > 2\ Q5: In the experiments, the score is chosen to be an L2 norm (see, for example, step 9 of Algorithm 1). What are the implications of this choice on the results? For example, are the results very different if we use another norm?
> >
> > My question related more to the length of the sets obtained (efficiency). Is there a real difference between the sizes of sets when using as a score function an l2 or l1 norm?

---

> > > ### Author Response · Authors · 2023-11-22
> > >
> > > Thank you for the clarification!
> > >
> > > > (Q2): It seems to me that it is never clearly stated in the paper that the scores are continuous (the choice of the l2 norm as the score function is presented as an example). I think this should be indicated in the theoretical statements (e.g. in lemma A.2).
> > >
> > > I see. In the background section between equations (1) and (2), we define the nonconformity score as $ A:
> > > \mathcal{Z}^{|D_{\text{train}} |} \times \mathcal{Z} \rightarrow \mathbb{R} $. We have updated the paper again to include this definition into the statement of Lemma A.2.
> > >
> > >
> > > > (Q5): In the experiments, the score is chosen to be an L2 norm (see, for example, step 9 of Algorithm 1). What are the implications of this choice on the results? For example, are the results very different if we use another norm?
> > >
> > > > My question related more to the length of the sets obtained (efficiency). Is there a real difference between the sizes of sets when using as a score function an l2 or l1 norm?
> > >
> > > The choice of nonconformity score is orthogonal to the main results of our paper, as we showed that we improve efficiency or validity over the CP baselines for *any* choice of nonconformity score. The implication of the choice is which quantity the algorithm would certify - e.g. do we want to bound the L1 or L2 error of the predictions? In our task of predicting 2/3d trajectories, we made the decision that a ball shaped confidence area (L2 norm) would make more practical sense than a diamond shaped one (L1 norm).
> > >
> > > That said, to answer your question, we ran the experiment with L1 on the particle simulation datasets. We report the mean of 3 trials. It is observable that in this experiment L1 is more efficient when there is less noise, and less efficient when the uncertainty is more pronounced.
> > >
> > >
> > > | particle ($\sigma=.01$)  | CFRNN | CFRNN |  CopulaCPTS | CopulaCPTS |
> > > |-----------------|-----------------|-----------------|-----------------|-----------------|
> > > |   |L2  |  L1 | L2 | L1  |
> > > | Cov (90%)    | 97.3   | 97.7   | 91.3   | 90.1  |
> > > | Area $\downarrow$    | 1.97    | 1.66    | 1.08    | 0.92    |
> > >
> > >
> > > |  particle ($\sigma=.05$) | CFRNN | CFRNN |  CopulaCPTS |CopulaCPTS |
> > > |-----------------|-----------------|-----------------|-----------------|-----------------|
> > > |   |L2  |  L1 | L2 | L1  |
> > > | Cov (90%)    | 94.5   | 95.0 | 90.6    | 90.0  |
> > > | Area $\downarrow$    | 5.80    | 6.72    | 5.27   | 6.15    |

---

### Official Review · Reviewer_3Gfg · 2023-11-06

**Soundness:** 2 fair
**Presentation:** 3 good
**Contribution:** 2 fair
**Rating:** 6
**Confidence:** 4

**Summary:**

The authors have proposed an extension of classical conformal prediction for multi-step, multivariate time series forecasting. Their work builds upon the foundations laid by Messoudi et al. without implying a significant improvement.

**Strengths:**

(1) The authors employ copulas to capture the relationships between different time steps in their time series analysis.

(2) Based on the experimental results, their method demonstrates a narrower predicted region compared to other approaches, suggesting enhanced efficiency.

**Weaknesses:**

(1) The primary weakness of this paper lies in its novelty. The use of copulas to capture uncertainty for multi-output scenarios is not a new concept. Notably, Messoudi et al. have previously proposed a similar approach, even employing the same empirical copula. While the authors assert that their contribution is the multivariate nature of every entry, it's important to note that the nonconformity score remains scalar. Therefore, the process of conformal prediction remains largely the same as in the univariate case.

(2) Another notable issue pertains to data type. While this paper primarily focuses on time series, upon closer examination, it becomes apparent that the temporal aspect of the data may not be as critical as initially assumed. The dataset comprises numerous time series, and each data point within these series appears to be exchangeable with one another. This exchangeability is what enables the application of conformal prediction. Furthermore, it's crucial to differentiate this work from other studies that delve into conformal prediction methods that extend beyond exchangeability.

**Questions:**

Q1: What distinguishes the proposed method from Messoudi's work, apart from the introduction of multivariate entries?

Q2: In comparison to another work[1] focusing on the multi-output version of conformal prediction that uses quantiles and can be extended to the multivariate case with the same score function, what distinguishes and contributes to this paper? I also recommend adding [1] as another baseline in the experimental part to prove the efficiency improvement.

Q3: In Equation 8, when aiming to enhance efficiency by minimizing the L1 norm of u, what is the rationale for choosing this particular norm?

Q4: Is there any typo in the loss function in the B.2 part? The loss function appears to be incorrect. Additionally, in the appendix, both equation (16) and equation (19) contain typos. The proof details regarding the validity (appendix A) are crucial; it would be beneficial to double-check this part.

[1] Feldman, S., Bates, S., & Romano, Y. (2023). Calibrated multiple-output quantile regression with representation learning. Journal of Machine Learning Research, 24(24), 1-48.

Following a thorough review of the authors' response and considering the feedback from other reviewers, I have decided to adjust the score to 6.

---

> ### Author Response · Authors · 2023-11-18
> **Response (part 1)**
>
> We thank the reviewer for providing helpful feedback.
>
> In response to the weaknesses (W) and questions (Q):
>
> > Q1/W1: What distinguishes the proposed method from Messoudi's work, apart from the introduction of multivariate entries?
>
> Our copula-based method is significantly different from Messoudi et al.  Our contribution is threefold:
> - **the two-step approach (section 4.1)**: Messoudi’s algorithm uses the same dataset to calibrate the individual scores and the copula, which results in inaccuracy in the estimation. We improve the process by (1) using separate calibration datasets for the individual time steps, and (2) constructing conformal predictive distributions for each timesteps. Our method leads to un-confounded calibrations and allows us to prove validity.
> - **validity guarantee (Theorem 4.1 and appendix A)**: In their paper, Messoudi et al. only showed empirical results and didn’t provide theoretical support for their performance.  As we have shown in table 1 in the paper, directly applying Copula often results in invalidity, which is a fatal flaw because guaranteed coverage is the main appeal of using conformal prediction. Our algorithm, on the other hand, is provably valid.
> - **New algorithm for finding** $s^*$ **(Eq. 8, and appendix B.2)**:  Copula calibration in Messsoudi et al is implemented as a grid search (grows exponentially with the dimension of the copula)  or by assuming that all $\alpha_j$ are the same (sacrifices efficiency). We mitigate this by using SGD for optimization, improving search time from exponential time to constant time.
>
> In conclusion, our algorithm resolved the shortcomings of Messoudi et al (invalidity and slowness), and presented a practical, sound, and fast algorithm for the big-data time series setting that does not yet exist in the conformal prediction literature.
>
> --
>
> > W1.1 While the authors assert that their contribution is the multivariate nature of every entry, it's important to note that the nonconformity score remains scalar. Therefore, the process of conformal prediction remains largely the same as in the univariate case.
>
> Our algorithm is not the same as the univariate case. Our contribution lies in combining multiple conformal predictions (can be multivariate or not, yes, with a scaler nonconformity score) together while maintaining coverage for all and having good efficiency. We argue that the second part departs from canonical CP and is novel.
>
>
> --
>
>
> > W2: data type. While this paper primarily focuses on time series, upon closer examination, it becomes apparent that the temporal aspect of the data may not be as critical as initially assumed….
>
> It is true that our algorithm is not limited to time series data. However, our work is motivated by many practical problems in time series that (1) has many exchangeable time series sequences, and (2) requires a confidence region coverage guarantee over a long prediction horizon, as exemplified in our choice of real-world datasets, anonymized medical data and autonomous driving. In addition, the structure of time series allows us to expand our algorithm to the autoregressive prediction setting, as explored in Appendix B.4.
>
>
> --
>
>
> > W2.2: Furthermore, it's crucial to differentiate this work from other studies that delve into conformal prediction methods that extend beyond exchangeability.
>
> We differentiated our setting with the CP-beyond-exchangeability method explicitly in our related works section (paragraph 2 under conformal prediction in section 2).
>
>
> --
>
>
> > Q2: In comparison to another work [1] focusing on the multi-output version of conformal prediction that uses quantiles and can be extended to the multivariate case with the same score function, what distinguishes and contributes to this paper? I also recommend adding [1] as another baseline in the experimental part to prove the efficiency improvement.
>
> Feldman et al. ’s work [1]  on multivariate conformal quantile regression can be used in conjunction with our copula methods, hence not a direct baseline. Specifically, we can use them for the individual timesteps where the dimensionality is lower, and then use copula to jointly calibrate the time steps, to further improve efficiency.
>
> We have attempted to implement this idea during the rebuttal. However, [1] requires calculating a grid over the output space, which grows exponentially with the output dimension. The original paper only evaluated performance for data with response dimension fewer than 4. In a time series setting, however, it is usual to have long prediction horizons. Running [1]’s algorithm on our particle simulation with output dimension =  50 (25 time steps x 2d output) requires 1048576.00 GiB of GPU memory.

---

> ### Author Response · Authors · 2023-11-18
> **Response (part 2)**
>
> > Q3: In Equation 8, when aiming to enhance efficiency by minimizing the L1 norm of u, what is the rationale for choosing this particular norm?
>
> Our algorithms holds as long as the constraint of  $C_{empirical}(\mathbf{u}*) \geq 1-\alpha$ holds, and this constraint is independent of the minimization goal. Therefore, the choice of this loss function is not very critical. In our experiments, we found no significant difference between using L1 or L2 norms.  We provide results on our first two synthetic datasets as example:
>
> | Method   | Particle ($\sigma=.01$) |          | Particle ($\sigma=.05$) |          |
> |----------|-------------------------|----------|-------------------------|----------|
> |          | Coverage (90%)          | Area $\downarrow$ | Coverage (90%)          | Area $\downarrow$ |
> | L1-loss  | $91.3 \pm 1.5$          | $1.08 \pm 0.14$     | $90.6 \pm 0.7$          | $5.27 \pm 1.02$     |
> | L2-loss  | $91.1 \pm 0.9$          | $0.97 \pm 0.16$     | $90.6 \pm 0.6$          | $5.40 \pm 0.89$     |
>
>
> --
>
>
> > Q4. Typos.
>
> We have fixed the loss function in B.2; thank you for pointing it out.
> Can you please elaborate on what typos you are referring to in equation (16) and (19)?
>
> --
>
> We thank the reviewer again for helping us improve our paper, and hope that our responses have cleared up the ambiguities. If so, we will appreciate it greatly if you can increase the score; If not, please let us know if you have any further questions.

---

### Comment · Area_Chair_Rh8u · 2023-11-21
**Engaging in discussion with the authors**

Dear reviewers, we are approaching the end of the discussion period  (Nov 22) with the authors , please read the rebuttal and engage with authors to  discuss any further questions/ clarifications you may have,

Many thanks

AC

---

### Meta-Review · Area_Chair_Rh8u · 2023-12-09

**Metareview:**

The paper proposes CopulaCPTS that allows conformal prediction for multistep predictor with statistical guarantees of validity. The copula models the dependency between the multi-step predictions. The calibration follow a 2 steps process: Each individial steps is calibrated on a different calibration set, then relying on a notion of multivariate quantile, the copula of the multi steps is calibrated to obtain a validity guarantee. The paper shows that this process leads to a validity guarantee. The copula calibration step is performed using SGD.

The work was contrasted and compared to previous work of Messoudi et al and authors explained that their work was an improvement of this method in terms of validity and speed.

The time series aspect of the work versus just multi-variate with exchangeability assumption was raised by 3Gfg and ruzn, which is a valid point. Authors clarified that there is no independence assumption considered between time steps.

Another criticism raised by reviewer bjQ8 is that the method needs very large calibartion sets in this 2 step process which may not be available.

Reviewers pointed out to many typos and inconsistencies that were corrected in the rebuttal. Reviewer XEiW raised an important point regarding the particular definition of the multivariate quantile and if there are other definitions. Authors should discuss more this in the paper.

On the empirical validation of the method, multiple baselines were proposed by the reviewers and the opinion were mixed regarding the advantages of this method on previous work. Authors stressed out that the main advantages are the guarantees for validity and the efficiency.

All in all, it is a good paper with two major weaknesses in terms of large calibration sets and better empirical comparison to other baselines.

**Justification For Why Not Higher Score:**

empirical assessment could do a better job with adding other baselines, and more challenging real world datasets.

**Justification For Why Not Lower Score:**

Improving a known good method  ( Messoudi et al ) and backing it with validation result is a good contribution.

---

### Decision · Program_Chairs · 2024-01-16

Accept (poster)